# Standing genetic variation fuels rapid adaptation to ocean acidification

M.C. Bitter [1]*, L. Kapsenberg [2], J.-P. Gattuso [3,4] & C.A. Pfister [1]

Global climate change has intensified the need to assess the capacity for natural populations to adapt to abrupt shifts in the environment. Reductions in seawater pH constitute a conspicuous global change stressor that is affecting marine ecosystems globally. Here, we quantify the phenotypic and genetic modifications associated with rapid adaptation to reduced seawater pH in the Mediterranean mussel, *Mytilus galloprovincialis*. We reared a genetically diverse larval population in two pH treatments ($pH_T$ 8.1 and 7.4) and tracked changes in the shell-size distribution and genetic variation through settlement. Additionally, we identified differences in the signatures of selection on shell growth in each pH environment. Both phenotypic and genetic data show that standing variation can facilitate adaptation to declines in seawater pH. This work provides insight into the processes underpinning rapid evolution, and demonstrates the importance of maintaining variation within natural populations to bolster species' adaptive capacity as global change progresses.

[1] Department of Ecology and Evolution, University of Chicago, 1101 E. 57th St., Chicago, IL 60637, USA. [2] Department of Marine Biology and Oceanography, CSIC Institute of Marine Sciences, Passeig Marítim de la Barceloneta, 37-49, E-08003 Barcelona, Spain. [3] Laboratoire d'Océanographie de Villefranche, Sorbonne Université, CNRS, 181 chemin du Lazaret, 06230 Villefranche-sur-mer, France. [4] Institute for Sustainable Development and International Relations, Sciences Po, 27 rue Saint Guillaume, 75007 Paris, France. *email: mcbitter@uchicago.edu

A fundamental focus of ecological and evolutionary biology is determining if and how natural populations can adapt to rapid changes in the environment. Recent efforts that have combined natural population censuses with genome-wide sequencing techniques have shown that phenotypic changes due to abrupt environmental shifts oftentimes occur concomitantly to signatures of selection at loci throughout the genome[1–6]. These studies demonstrate the importance of standing genetic variation in rapid evolutionary processes[7], and challenge classical population genetic theory, which assumes that most genetic variation has a small effect on fitness and that selective forces alter this variation gradually over a timescale of millennia[8]. Though the mechanisms maintaining such significant levels of genetic variation in natural populations have been historically debated, recent population genomic surveys have begun to elucidate the role of spatially and temporally variable selection pressures in this process[8]. For example, environmental gradients in marine systems can maintain signatures of balanced polymorphisms between populations at multiple, putatively functional loci, even amidst high levels of gene flow[9–12]. While this variation has enabled the persistence of natural populations inhabiting the contemporary and historic regimes of environmental variability, it is unclear whether it will facilitate the magnitude and rate of adaptation necessary for species persistence under the conditions expected as a result of global climate change[13].

One pertinent threat facing marine species is ocean acidification, the global-scale decline in seawater pH driven by oceanic sequestration of anthropogenic carbon dioxide emissions[14]. The current rate of pH decline is unprecedented in the past 55 million years[15], and lab-based studies have shown negative effects of expected pH conditions on a range of fitness-related traits (e.g., growth, reproduction, and survival) across life-history stages and taxa[16]. Marine bivalves are one of the most vulnerable taxa to ocean acidification[17,18], particularly during larval development[19]. The ecologically and economically valuable Mediterranean mussel, *Mytilus galloprovincialis*, is an exemplary species for studying the effects of ocean acidification on larval development. Low-pH conditions reduce shell size and induce various, likely lethal, forms of abnormal larval development[20,21]. Sensitivity to low pH, however, can vary substantially across larvae from distinct parental crosses, suggesting that standing genetic variation could fuel an adaptive response to ocean acidification[21].

Here, we explored the potential for, and dynamics of, rapid adaptation to ocean acidification in *M. galloprovincialis*. We quantified the effects of low-pH exposure on phenotypic and genetic variation throughout development in a single population of *M. galloprovincialis* larvae (Fig. 1). Larvae were reared in ambient and low-pH and (i) shell-size distributions were quantified on days 3, 6, 7, 14, and 26; (ii) the frequency of 29,400 single nucleotide polymorphisms (SNPs) across the species' exome was estimated on days 6, 26, and 43; and (iii) signatures of selection on larval shell size were determined in each treatment. To generate a starting larval population representative of the standing genetic variation within a wild population of *M. galloprovincialis*, 16 males were crossed to each of 12 females, hereafter referred to as founding individuals ($N = 192$ unique crosses). The resulting larval population was reared in an ambient (pH$_T$ 8.05, $N = 6$ replicate buckets) and low-pH treatment (pH$_T$ 7.4–7.5, $N = 6$ replicate buckets). While the low-pH treatment falls outside the range of annual variability the population currently experiences (pH$_T$ ~7.8–8.1)[21], normal development of *M. galloprovincialis* larvae can occur at this pH[21]. We thus expected, a priori, that this value would effectively reveal the presence of variation underpinning low-pH tolerance. Furthermore, while our low-pH treatment falls below the expected 0.4 pH$_T$ unit decline in global mean seawater pH by 2100[14], marine species occupying unequilibrated coastal regions, such as the lagoon habitat of the study population, may periodically experience pH conditions that fall far below projected means during the next century[22].

Our results indicate substantial variation for low-pH tolerance within *M. galloprovincialis*, and demonstrate that genotypes exhibiting elevated fitness in ambient conditions are distinct from those exhibiting elevated fitness in low-pH conditions. In a broader framework, this study demonstrates a polygenic basis to a rapid adaptive response and suggests the importance of maintaining variation within natural populations to bolster species resilience as global change progresses.

## Results

**Phenotypic trajectories**. As expected, shell size was significantly affected by pH treatment throughout the experiment (likelihood ratio test, $p = 0.029$), and shell length of low-pH larvae was 8% smaller than that of larvae reared in ambient pH on days 3 and 7. Shell length was affected by the interaction of day and treatment (likelihood ratio test $p < 0.001$), indicating treatment-specific growth patterns. From days 7 to 26 the size distributions in each treatment began to converge, with larvae in low pH being only

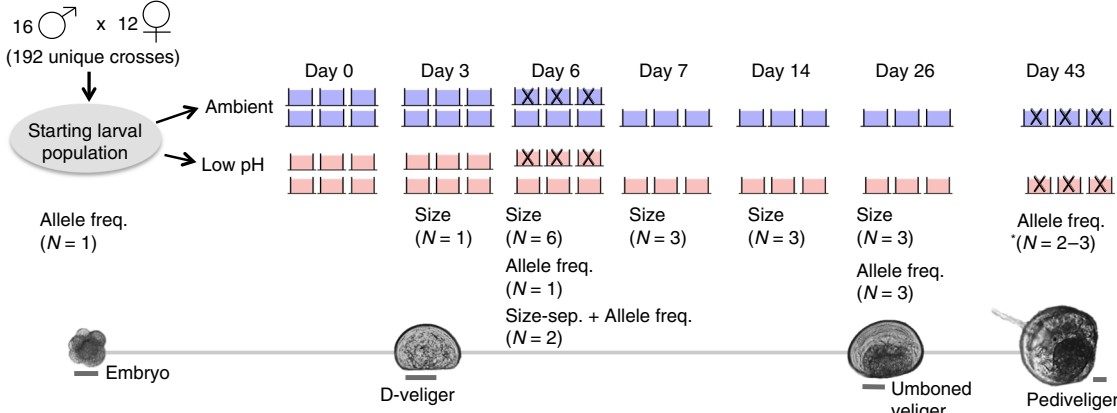

**Fig. 1 Experimental schematic.** Pictures of larvae at key developmental points, cross design, and replication and sampling strategy throughout the experiment. Scale bar for larval pictures set at 50 μm. Blue and red buckets correspond to ambient and low-pH treatments, respectively. Replicate buckets marked with an "X" were destructively sampled (i.e., all larvae removed/preserved) and thus absent from the experimental system on subsequent sampling days. *Allele frequency data from two replicate buckets in the ambient treatment was generated on day 43, as the third replicate bucket was sacrificed to optimize protocol for sampling settled larvae.

2.5% smaller than those cultured in the ambient treatment by day 26 (Fig. 2). In addition, treatment-specific patterns of phenotypic variation were observable during this period. Specifically, the coefficient of variation of shell size was elevated in the low-pH treatment on day 3 (ambient pH CV = 3.52; low-pH CV = 5.16) and day 7 (ambient pH CV = 3.54; low-pH CV = 4.98). This difference in phenotypic variation was reduced on day 14 (ambient pH CV = 8.83; low-pH CV = 9.7) and no longer evident by day 26 (ambient pH CV = 11.66; low-pH CV = 11.26).

**Changes in genetic variation.** We identified 29,400 SNPs across the species exome that were present within the larval population across all sampling days and treatments. To link the observed phenotypic trends in each treatment to changes in this variation, we analyzed the SNP's using principal component analysis (PCA), outlier loci identification, and a statistical metric of genomic differentiation ($F_{ST}$). For both ambient and low-pH treatments, all analyses indicated increasing genomic differentiation of the larval cultures away from the day 0 larval population. This trend is visually apparent in the PCA, which incorporated allele frequency data from all larval samples collected during the pelagic stage and settlement (excluding size-separated groups) (Fig. 3). Through time there is an observed increase in Euclidian distance among samples (e.g., days 26 and 43). This may be driven, at least in part, by selection-induced

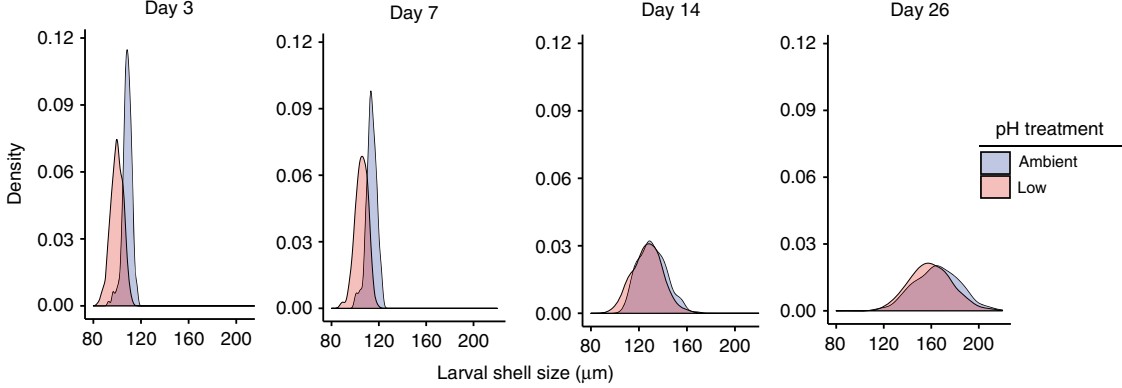

**Fig. 2 Larval size distributions throughout the shell growing period.** Blue and red densities correspond to shell-size distributions in ambient and low-pH treatments, respectively. Larval size was significantly affected by treatment (likelihood ratio test, $p = 0.029$) and the interaction of day and treatment (likelihood ratio test, $p < 0.001$) throughout the shell growing period. Source data for this figure are provided in the Source Data file.

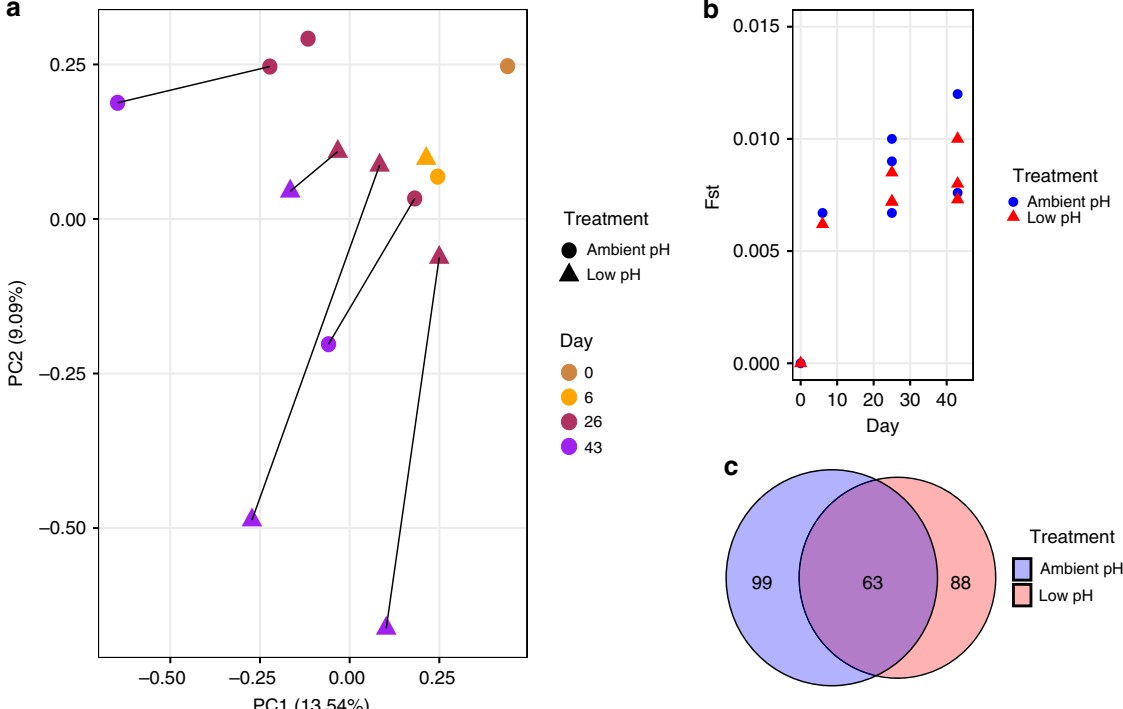

**Fig. 3 Patterns of genomic variation within larval populations throughout development. a** Principal component analysis of allele frequency data from larval samples collected throughout the course of the experiment. Allele frequency data from 29,400 SNPs were used for PCA. PCA point color corresponds to developmental day, while shape corresponds to treatment condition. **b** $F_{ST}$ between the day 0 larval population and each larval population treatment replicate throughout development. Blue circles and red triangles correspond to ambient and low pH replicate bucket $F_{ST}$ values, respectively. **c** Venn diagram representing the extent of overlap in outer loci identified in ambient (blue) and low-pH (red) treatments. Source data for this figure are provided in the Source Data file.

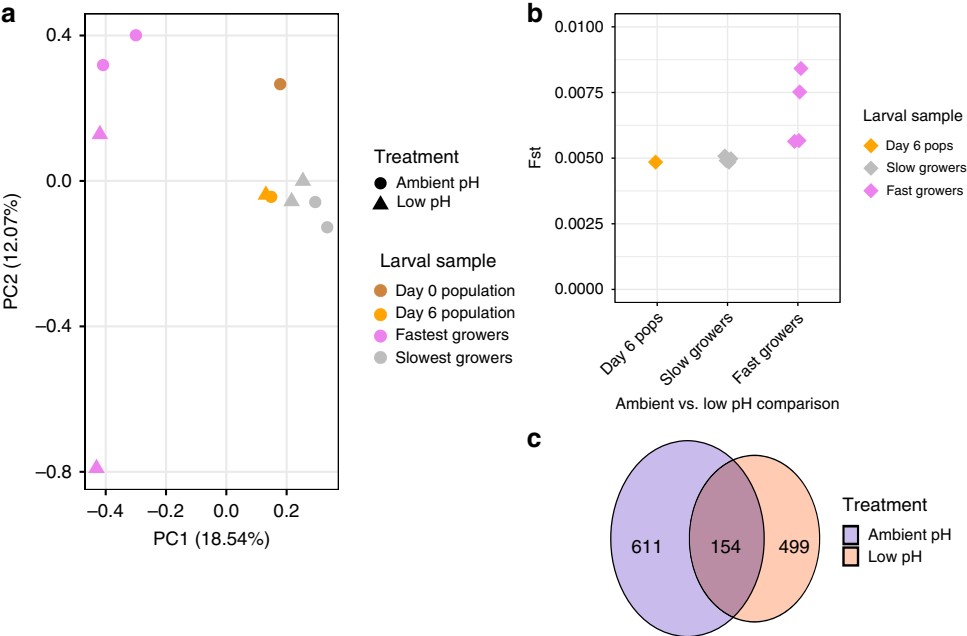

**Fig. 4 Patterns of selection on shell growth in ambient and low-pH conditions. a** Principle component analysis examining genomic signature of size separation in ambient and low-pH treatments. Allele frequency data are based on 29,400 SNPs and all samples collected on day 6, as well as the day 0 starting larval population. PCA point color corresponds to developmental day, while shape corresponds to treatment. **b** Exome-wide $F_{ST}$ computed pairwise between ambient and low-pH replicate buckets for the entire larval population (day 6 pops) and the size-selected larvae isolated on day 6 (triangle color corresponds to larval sample) (day 6 Pops: $N = 1$ pairwise comparison; Slowest and Fastest Growers: $N = 4$ pairwise comparisons). **c** Venn diagram representing extent of overlap in loci displaying signatures of selection for shell growth in ambient (blue) and low-pH (red) treatments. Source data for this figure are provided in the Source Data file.

declines in larval survival throughout the pelagic phase, as well as an increase in the influence of allele frequency drift among replicate buckets (i.e., persistent random mortality leading to replicate-specific patterns of selection). Observations of sustained larval mortality throughout the experiment (indicated via empty D-veliger shells in buckets) corroborated these trends, as did our gross estimate of mortality between days 0 and 26 (97.7% and 97.6% in ambient and low-pH treatments, respectively).

We identified SNPs that changed significantly in frequency between the day 0 larval population and the larvae sampled on days 6, 26, and 43 in each treatment. Significant SNPs were identified using a rank-based approach and the observed allele frequency shift probabilities generated from the Fisher's Exact and Cochran–Mantel–Haenszel (CMH) tests. This analysis indicated pervasive signatures of selection in both treatments, with thousands of SNPs significantly changing in frequency throughout the course of the experiment (Supplementary Fig. 1). To identify a list of outlier loci for each treatment, we identified loci containing significant SNPs on all sampling days, thus leveraging independent replicate cultures and multiple sampling days (see "Methods" section for details). This resulted in the identification of 162 and 151 total outlier loci in the ambient and low-pH conditions, respectively. We compared the overlap in these lists to identify pH-specific outliers (loci that were outliers in only one of the pH treatments) and shared outliers (loci that were outliers in both pH treatments). In total, we identified 99 ambient pH-specific outlier loci (31 annotated), 88 low pH-specific outlier loci (29 annotated), and 63 shared loci (24 annotated) based on transcriptome provided in Moreira et al.[23] (see Supplementary Data 1 for outlier annotations and Supplementary Data 2–4 for outlier loci sequences). Therefore, 58% of the outlier identified in the low-pH treatment were unique to that environment (i.e., did not display strong signatures of selection in the ambient treatment) (Fig. 3c). This finding highlights the

potential polygenic nature of low-pH adaptation and demonstrates that natural populations currently harbor variation at these putatively adaptive loci.

Another statistical metric of genetic differentiation, $F_{ST}$, was used to identify changes in the magnitude of selection throughout development. We computed exome-wide estimates of $F_{ST}$ pairwise between the day 0 larval population and each available replicate bucket on all sampling days. The greatest change in $F_{ST}$ occurred between days 0 and 6, before elevating more slowly thereafter, suggesting that the majority of selective mortality in *M. galloprovincialis* larvae occurred prior to day 6 (Fig. 3b).

**Signatures of selection on shell growth in ambient and low pH**. The size separation of larvae on day 6 isolated the largest 18% from the smallest 82% of shell growers in the ambient, and the largest 21% from the smallest 79% of shell growers in the low-pH treatment (Supplementary Fig. 2). Hereafter, these groups will be referred to as the fastest and slowest growers, respectively. Shell size on day 6 was significantly affected by treatment and size class (likelihood ratio test, $p < 0.001$). PCA using allele frequency data from the day 0 starting larval population and larval samples collected on day 6 revealed a strong genetic signature of size class (Fig. 4a). Specifically, the fastest growers segregated along PC1 from the slowest growers in both treatments, with the day 0 larval population and day 6 larval population samples (from each treatment) falling in between the size-separated groups. The number of significant SNP's differentiating the fastest and slowest growers in each treatment, hereafter referred to as size-selected SNPs, was comparable: 963 significant SNPs were identified in ambient and 846 significant SNPs were identified in the low-pH treatment (significance determined using CMH test). This led to the identification of 611 size-selected loci that were unique to the ambient pH treatment (225 annotated), 499 size-selected loci that

were unique to the low-pH treatment (184 annotated), and 154 size-selected loci (51 annotated) that were shared between environments (Supplementary Data 5 contains existing annotations for size-selected genes in each treatment and Supplementary Data 6–8 contain associated sequences for size-selected genes). Therefore, 76% of loci associated with fast shell growth in low-pH were not associated with fast growth in the ambient treatment, indicating unique targets of selection on shell growth in each environment (Fig. 4c). $F_{ST}$ analysis corroborated this trend, as elevated signatures of differentiation were observed between the fastest growers in ambient and low pH, relative to differentiation between the slowest growers and full larval populations in ambient and low-pH treatments (Fig. 4b).

## Discussion

While previous work has shown strong negative effects of low pH on larval development in bivalves[17–19,24], the results presented here suggest that standing variation within the species could facilitate rapid adaptation to ocean acidification. Observed shell length differences on days 3 and 7 matched expectations for the species based on previous work (−1 μm per 0.1 unit decrease in pH)[21]. However, this difference was reduced ~50% by day 14. Mechanistically, low-pH treatment effects on bivalve larval shell growth are driven by the limited capacity of larvae to regulate carbonate chemistry, specifically aragonite saturation state, in their calcifying space[25–27]. Our data show that this physiological limitation is greatest prior to day 7, after which a partial convergence of the size distributions in the ambient and low-pH treatments was observed.

It is likely that the partial convergence of shell-size distributions observed by day 14 was, at least in part, driven by natural selection for low-pH tolerance. We have previously shown that the smallest D-veligers in the low-pH treatment display an increased prevalence of morphological abnormalities, which likely become lethal during the shell growth period[21]. Directional selection against this phenotypic group would shift the size distribution closer to that of larvae reared in ambient conditions, as we observed. The unique signatures of selection on shell growth after 6 days of exposure to low pH, as well as the unique outlier loci identified in the low-pH environment throughout the larval period, further strengthen the notion that these phenotypic trends were rooted in changes in the larval population's underlying genetic variation.

Documentation of substantial mortality by day 26 and increasing genetic differentiation throughout the experiment, as evidenced by PCA, significant changes in SNP frequencies, and increasing values of $F_{ST}$, further suggest the process of selection during the shell growth period. The observed trends in $F_{ST}$, however, highlighted a developmental point of heightened selection occurring before the convergence of the shell-size distributions. Specifically, when $F_{ST}$ is scaled by duration of treatment exposure, the genomic differentiation between the day 0 larval population and the larval populations on day 6 was three and five times greater than that observed between the day 0 larval population and the larval populations on days 26 and 43, respectively. This suggests that a largely singular, intense selection event occurred prior to day 6 and may be responsible for the majority of genetic differentiation that occurs during larval growth and settlement. We recently identified two specific early developmental processes that are sensitive to low-pH conditions and occur in this timeframe[21]. These processes include the formation of the shell field (early trochophore stage) and the transition between growth of the first and second larval shell (late trochophore stage), both of which occur within 48 h of fertilization and result in a suite of size-dependent morphological abnormalities that likely become lethal during the shell growth period[21]. Traditionally, metamorphosis from the swimming D-veliger to the settled juvenile is regarded as the main genetic bottleneck during the development of marine bivalve larvae[28]. Our sampling from the embryo stage through settlement, however, suggests that there is a major selection event prior to day 6 that may have an even larger effect on shaping genotypes of settled juveniles than any selection thereafter.

Additional factors that may have led to the observed phenotypic dynamics are food-augmented acclimation and selective mortality via food competition. It has been demonstrated that increased energy availability can allow marine invertebrates to withstand pH stress[29] and, in the case of Mytilus edulis, food availability can mitigate the negative effects of ocean acidification[30]. This compensation, however, is unlikely in our experiment. Our algal concentrations during the period of phenotypic convergence (days 7–26) fell below optimal concentrations reported for the species[31,32]. In addition, settlement was not observed in our experiment until 40 days into development. This falls outside the 3–5 week larval pelagic phase documented in previous work[31–33], and further demonstrates that the larvae were indeed food limited in each treatment. This food limitation may have induced intraspecific competition and facilitated the selective mortality of less fit genotypes, thereby producing the pervasive signatures of selection observed in both treatments starting on day 6. Furthermore, selection may have been concentrated on the smallest larvae in low pH, thus driving the phenotypic convergence between treatments. Ultimately, surviving larvae in the low-pH treatment were able to partially compensate for the negative effect of $CO_2$-acidification on calcification kinetics[26]. As pH tolerance has been shown to exhibit heritability in Mytilus spp.[34,35], it is possible that multigenerational selection may indeed allow the population to ultimately recover the offset in shell-size observed in this experiment.

We identified hundreds of SNPs responding to each pH treatment throughout the larval period and, via stringent filtering techniques, further winnowed these candidates to identify 162 outlier loci in ambient and 151 outlier loci in low-pH conditions. Eighty-eight of the low-pH outliers (58%) were statistically unchanged in the ambient conditions. While some of this treatment disparity may be an artifact (i.e., false positives in the low pH or false negatives in the ambient treatment), it is unlikely that this is the case for all the unique outliers identified, given our stringent filtering techniques. These data thus provide evidence that loci that were not critical for fitness in the ambient environment came under strong selection in the low-pH environment. The limited amount of overlap in size-selected loci and elevation of $F_{ST}$ differentiation between the fastest growers further indicate novel targets of selection for accelerated shell growth in the low-pH environment. As shell growth is a direct proxy for fitness[21,36], these data suggest that the most fit genotypes in ambient conditions may not be the individuals that harbor the adaptive genetic variation necessary to improve fitness in simulated ocean acidification. Furthermore, the near-equal magnitude of mortality observed in each treatment suggests that adaptation to expected declines in seawater pH within this species may proceed via a reshuffling of standing genetic variation, effectively allowing populations to bypass the dramatic declines in population size that are characteristic of rapid evolution to abrupt environmental shifts via fixation of novel mutations[13].

Ultimately, these findings contribute to an emerging body of work demonstrating that rapid adaptation can indeed exhibit a polygenic basis[2–4,37]. Variation at functional loci across the genome is likely maintained in populations via multilocus balancing selection between populations spanning environmental gradients[9–12], or within populations experiencing temporally

fluctuating selection pressures[3,38]. For example, populations of *M. edulis* inhabiting the Baltic Sea are periodically exposed to the pH conditions used in this study[34]. Given extensive gene flow and hybridization of *Mytilus* populations throughout the Baltic Sea, Atlantic, and Mediterranean, it is possible such gradients could maintain adaptive low-pH tolerance throughout these regions, though we lack explicit demonstration of this within the focal population.

The dynamics observed in this study further suggest a potential role of cryptic genetic variation (CGV) in adaptation to climate change. CGV is defined as a subclass of standing variation with a conditional effect, such that it becomes adaptive during evolution to rare or novel environmental conditions[39]. This conditional effect may manifest as a genotype-by-genotype interaction, in which an allele's effect is conditional upon the genetic background (resulting from dominance or epistasis), or genotype-by-environment interactions, in which an allele's effect is conditional upon the environment[39]. In the context of the present study, the low-pH treatment value is indeed rare in the study population's natural habitat[21] and the pH-specific signatures of selection indicate some of the variation that was putatively neutral in the ambient treatment were conditionally beneficial in low-pH. Furthermore, a defining characteristic of adaptation via cryptic variation is the release of phenotypic variation in the novel environment[39]. Accordingly, we observed elevation of phenotypic variation in the low-pH treatment on days 3 and 7, coinciding with the inferred period of heightened selection (between days 0 and 6). Kingston et al.[35] similarly showed that stressful conditions (low pH, high temperature, and low food conditions) released phenotypic variation in calcification rates in two species of *Mytilus* mussels, and went on to link this variation to a number of loci of moderate effect[35]. While an important avenue of future research is determining which CGV mechanisms (genotype-by-genotype or genotype-by-environment interactions) may be producing such patterns, the economic and ecological importance of marine mussels, as well as their global exposure to declining seawater pH, highlight the need to conserve standing variation in order to allow the adaptive capacity of natural populations to play out as climate change progresses. Furthermore, exploring the interplay of standing variation maintained by balanced polymorphisms and cryptic variation during adaptation to climate change is an exciting avenue of future research, which may lend fundamental insights into the dynamics of rapid evolutionary processes.

Our list of low-pH outlier loci provide targets of natural selection as ocean acidification progresses. Notably, this list included an *HSPA1A* gene, which encodes heat shock protein 70 (HSP70) (NCBI Accession: XM_022468949), one of a group of gene products whose expression is induced by physiological stressors and generally work to mediate/prevent protein denaturation and folding[40]. While substantial evidence has documented the role of *HSP70* in the thermal stress response across a range of taxa[40], emerging transcriptomic studies have also demonstrated the protein's role in the physiological response to low-pH conditions in marine bivalves[41].

The influence of ocean acidification conditions on the biomineralization process of calcifying marine species has been readily documented[16]. Indeed, two of our candidate low-pH outliers mapped to a tyrosinase (NCBI Accession: XM_022487312) and chitinase (NCBI: Accession: MG827131.1) gene, each part of pathways known to be involved in calcium carbonate shell formation in marine bivalves[42]. Tyrosinase genes most broadly function in the process of sclerotization, the mechanism by which marine bivalves form the shell's periostracum[43], and activity of tyrosinase is heightened during larval shell biogenesis[44,45]. The outlier identified in this study mapped to a *tyrosinase tyr3*

isoform, which has previously been shown to be upregulated in response to low-pH conditions in the Antarctic pteropod[46] and *M. edulis*[47]. Chitinase genes produce organic scaffolds during shell formation in marine bivalves and have additionally been shown to change expression in response to seawater acidification in *Mytilus*[47].

While such gene expression-based studies provide insight into the underlying physiological responses to changes in seawater chemistry, our study demonstrates the presence of underlying genetic variation within these putatively adaptive loci. This provides, to our knowledge, the first documentation of standing genetic variation at functionally relevant loci within marine bivalves, and ultimately offers robust evidence for the species' capacity to adapt to changes in seawater pH. We are currently investigating these candidates more deeply through a combination of comparative transcriptomics, quantitative PCR and in situ hybridizations.

Many outlier loci in low pH were also outliers in the ambient treatment (42%). This likely represents the action of selection against recessive homozygotes within the population, termed genetic load[48], and selection induced by the laboratory regime. The influence of genetic load has been demonstrated to induce signatures of selection in neutral environments in a range of highly fecund species, such as plants and marine bivalves[28,49]. Our crossing scheme likely amplified this signature, as equal proportions of all pairwise crosses were conducted, thereby maximizing the likelihood of lethal, or less fit, homozygotes in the day 0 larval population. These shared signatures of selection could be further associated with selective pressures induced by the laboratory conditions, such as salinity, temperature, or the food resources, which are independent of the pH manipulation, yet still may have favored a subset of genetic backgrounds in this genetically diverse species[50].

Species persistence as global climate change progresses will, in part, hinge upon their ability to evolve in response to the shifting abiotic environment[13]. Our data suggest that the economically and ecologically valuable marine mussel, *M. galloprovincialis*, harbors standing variation that would facilitate rapid adaptation to ocean acidification. We have further demonstrated that genotypes exhibiting elevated fitness in current ocean conditions may be distinct from those exhibiting elevated fitness in future oceans. Ultimately, these findings support conservation efforts aimed at maintaining variation within natural populations to increase species resilience to future ocean conditions. In a broader evolutionary framework, the substantial levels of genetic variation present in natural populations have historically puzzled evolutionary biologists[51]. Though this study does not address the processes that maintain this variation, we demonstrate its utility in rapid adaptation, thereby advancing our understanding of the mechanisms by which natural populations evolve to abrupt changes in the environment.

## Methods

**Larval cultures**. Mature *M. galloprovincialis* individuals were collected in September 2017 from the underside of a floating dock in Thau Lagoon (43.415 °N, 3.688 °E), located in Séte, France. Thau Lagoon has a mean depth of 4 m and connection to the Mediterranean Sea by three narrow channels. pH variability at the collection site during spawning season ranges from $pH_T$ 7.80 to 8.10[21]. Mussels were transported to the Laboratoire d'Océanographie (LOV) in Villefranche-sur-Mer, France and stored in a flow-through seawater system maintained at 15.2 °C until spawning was induced.

Within 3 weeks of the adult mussel collection, individuals were cleaned of all epibiota using a metal brush, byssal threads were cut, and mussels were warmed in seawater heated to 27 °C (~+12 °C of holding conditions) to induce spawning. Individuals that began showing signs of spawning were immediately isolated, and allowed to spawn in discrete vessels, which were periodically rinsed to remove any potential gamete contamination. Gametes were examined for viability and stored on ice (sperm) or at 16 °C (eggs). In total, gametes from 12 females and 16 males

were isolated to generate a genetically diverse starting larval population. To produce pairwise crosses, 150,000 eggs from each female were placed into 16 separate vessels, corresponding to the 16 founding males. Sperm from each male was then used to fertilize the eggs in the corresponding vessel, thus eliminating the potential effects of sperm competition and ensuring that every male fertilized each female's eggs. After at least 90% of the eggs had progressed to a four-cell stage, equal volumes from each vessel were pooled to generate the day 0 larval population (~2 million individuals), from which the replicate culture buckets were seeded. A total of 100,000 individuals were added to each culture buckets ($N = 12$, 18 embryos mL$^{-1}$). The remaining embryos were frozen in liquid nitrogen, and stored at $-80\,°C$ for DNA analysis of the day 0 larval population. Likewise, gill tissue was collected from all founding individuals and similarly stored for downstream DNA analyses. Larvae were reared at $17.2\,°C$ for 43 days. Starting on day 4, larvae were fed $1.6 \times 10^8$ cells of *Tisochrysis lutea* daily. Beginning on day 23, to account for growth and supplement diet, larvae diet was complemented with 0.2 µL of 1800 Shellfish Diet (Reed Mariculture) (days 23–28 and day 38) and ~$1.6 \times 10^8$ cells of *Chaetoceros gracilis* (days 29–37 and 39–41). Algae were added as a pulse to each experimental replicate, twice daily (early morning, late afternoon). As the system was flow-through on days 0–26, the density of algae declined between consecutive pulses. The number of algae in each pulse was determined daily within the algal stock solution, and values reported above are the average of daily algal additions. Evidence of food consumption by larvae was indicated by observed food in larval guts, as well as substantial growth of larvae throughout the experiment.

**Larval sampling**. We strategically sampled larvae throughout the experiment to observe phenotypic and genetic dynamics across key developmental events, including the trochophore to D-veliger transition (day 6), the shell growth period (days 4–26), and the metamorphosis from D-veligers to settlement (days 40–43). On day 6 of the experiment, larvae were sampled from three of the six replicate buckets per treatment. A subset of larvae ($N = 91$–172) from each bucket was isolated to obtain shell length distributions of larvae reared in the two treatments. The remaining larvae were separated by shell size using a series of six Nitex mesh filters (70, 65, 60, 55, 50, and 20 µm; Supplementary Fig. 2b) and frozen at $-80\,°C$. The smallest size group contained larvae arrested at the trochophore stage, and therefore unlikely to survive. The remaining five size classes isolated D-veligers from the smallest to the largest size. The shell length distribution of the larvae was used to inform, a posteriori, which combination of size classes would produce groups of the top 20% and bottom 80% of shell growers from each treatment. The relevant size groups from two replicates per treatment were then pooled for downstream DNA analysis of each phenotypic group. For the third replicate, a posteriori, all size groups were pooled in order to compute the allele frequency distribution from the entire larval population in each treatment on day 6. This sample was incorporated into analyses of remaining replicate buckets, which were specifically used to track shifts in phenotypic and genetic dynamics throughout the remainder of the larval period in each treatment.

Following size separation on day 6, the remaining replicate buckets ($N = 3$ per treatment) were utilized to track changing phenotypic and allele frequency distributions in the larval population through settlement. Larvae were sampled for size measurements on day 3 ($N = 30$–36 individuals), day 7 ($N = 38$–71 individuals), day 14 ($N = 37$–104 individuals), and day 26 ($N = 49$–112 individuals). Also on day 26, an additional ~1,000 larvae per replicate were frozen and stored at $-80\,°C$ pending DNA analysis. Finally, on day 43, settled individuals were sampled from each bucket (settlement was first observed on day 40 in all buckets). Treatment water was removed, and culture buckets were washed three times with FSW to remove unsettled larvae. Individuals that remained attached to the walls of the bucket were frozen and stored at $-80\,°C$ for DNA analysis.

**Culture system and seawater chemistry**. Larvae were reared in a temperature-controlled sea table ($17.2\,°C$) and 0.35 µm filtered and UV-sterilized seawater (FSW), pumped from 5 m depth in the bay of Villefranche. Two culture systems were used consecutively to rear the larvae, both of which utilized the additions of pure $CO_2$ gas for acidification of FSW. First, from days 0 to 26 the larvae were kept in a flow-through seawater pH-manipulation system. Briefly, seawater pH (pH$_T$ 8.05 and pH$_T$ 7.4) was controlled in four header tanks using a glass pH electrode feedback system (IKS aquastar) and pure $CO_2$ gas addition and constant $CO_2$-free air aeration. Two header tanks were used per treatment to account for potential header tank effects. Each header tank supplied water to three replicate culture buckets (drip rate of 2 L h$^{-1}$), fitted with a motorized paddle and Honeywell Durafet pH sensors for treatment monitoring (see Kapsenberg et al.[52] for calibration methods).

On day 27 of the experiment, the flow-through system was stopped due to logistical constraints and treatment conditions were maintained, in the same culture buckets, using water changes every other day. For water changes 5 L of treatment seawater (70% of total volume) was replaced in each culture using FSW preadjusted to the desired pH treatment. Seawater pH in each culture bucket was measured daily, and before and after each water change.

All pH measurements (calibration of Durafets used from days 0 to 26 and monitoring of static cultures from days 27 to 43) were conducted using the spectrophotometric method and purified *m*-cresol dye and reported on the total scale (pH$_T$)[53]. Samples for total alkalinity ($A_T$) and salinity were taken from the

header tanks every 2–3 days from days 0 to 26 and daily during the remainder of the experiment. $A_T$ was measured using an open cell titration on Metrohhm Titrando 888[53]. Accuracy of $A_T$ measurements was determined using comparison to a certified reference material (Batch #151, A. Dickson, Scripps Institution of Oceanography) and ranged between $-0.87$ and $5.3$ µmol kg$^{-1}$, while precision was 1.23 µmol kg$^{-1}$ (based on replicated samples, $n = 21$). Aragonite saturation and $pCO2$ were calculated using pH and $A_T$ measurements and the *seacarb* package[54] in R with dissociation constants K$_1$ and K$_2$[55], Kf[56] and Ks[57]. Seawater chemistry results are presented in Supplementary Tables 1 and 2.

**Shell-size analysis**. Shell size was determined as the maximum shell length parallel to the hinge using brightfield microscopy and image analysis in ImageJ software. All statistical analyses were conducted in R (v. 3.5.3). As larval shell length data did not pass normality tests (Shapiro–Wilk test), shell-size was log-transformed to allow parametric statistical analysis. We tested the effect of day, treatment, and the interaction of the two using linear-mixed effects models, with day and treatment as fixed effects and replicate bucket as a random effect (*lmer*). Effects of treatment and size class on log-transformed shell length from size-separated larvae were also analyzed using a linear-mixed effect model in which size class, treatment, and their interaction were fixed effects, while larval bucket was a random effect. Significance of the fixed effects were tested against a null model using a likelihood ratio test.

**DNA extraction and exome sequencing**. We implemented exome capture, a reduced-representation sequencing approach, to identify SNPs and their frequency dynamics throughout the course of the experiment. Exome capture targets the protein-coding region of the genome, and thus increases the likelihood that identified polymorphisms are in or near functional loci[58]. Genomic DNA from each founding individual and larval sample was extracted using the EZNA Mollusc Extraction Kit, according to manufacturer's protocol. DNA was quantified with a Qubit, and quality was determined using agarose gel, Nanodrop (260/280), and TapeStation analysis.

Genomic DNA was hybridized to a customized exome capture array designed and manufactured by Arbor Biosciences (Ann Arbor, Michigan) and using the species transcriptome provided in Moreira et al.[23]. Specifically, in order to design a bait set appropriate for capture of genomic DNA fragments, 90-nucleotide probe candidates were tilled every 20 nucleotides across the target transcriptome contigs. These densely tiled candidates were MEGABLASTed to the *M. galloprovincialis* draft genome contigs available at NCBI (GCAA_001676915.1_ASM167691v1_genomic.fna), which winnowed the candidate list to only baits with detected hits of 80 nucleotides or longer. After predicting the hybrid melting temperatures for each near-full-length hit, baits were further winnowed to those with at most two hybrids of $60\,°C$ or greater estimated melting temperature in the *M. galloprovincialis* genome. This collection of highly specific baits with near-full-length hits to the draft genome were then down-sampled to a density of roughly one bait per 1.9 kbp of the final potential target space, in order to broadly sample the target while still fitting within our desired number of myBaits kit oligo limit. The final bait set comprises 100,087 oligo sequences, targeting 94,668 of the original 121,572 transcriptome contigs.

Genomic DNA from each sample was subject to standard mass estimation quality control, followed by sonication using a QSonica QR800 instrument and SPRI-based dual size selection to a target modal fragment length of 350 nucleotides. Following quantification, 300 ng total genomic DNA was taken to library preparation using standard Illumina Truseq-style end repair and adapter ligation chemistry, followed by six cycles of indexing amplification using unique eight nucleotide dual index primer pairs. For target enrichment with the custom myBaits kit, 100 ng of each founder-derived library were combined into two pools of 14 libraries each, whereas 450 ng of each embryonic and larval-pool derived library were used in individual reactions. After drying the pools or individual samples using vacuum centrifugation to 7 µL each, Arbor followed the myBaits procedure (v. 4) using the default conditions and overnight incubation to enrich the libraries using the custom probe set. After reaction cleanup, half (15 µL) of each bead-bound enriched library was taken to standard library amplification for ten cycles using Kapa HiFi polymerase. Following reaction cleanup with SPRI, each enriched library or library pool was quantified using qPCR, indicating yields between 30 and 254 ng each.

The captured libraries were sequenced at the University of Chicago Genomics Core Facility on three lanes of Illumina HiSeq 4000 using 150-bp, paired-end reads. The captured adult libraries were sequenced on an individual lane, while the 22, pooled larval samples were split randomly between the remaining two lanes. Average coverage for founding individuals was 40×, while average coverage in pooled samples was 100×.

**Read trimming and variant calling**. Raw DNA reads were filtered and trimmed using Trimmomatic[59], and aligned to the species reference transcriptome provided in Moreira et al.[23] using bowtie2[60]. Variants in the founding individuals were identified using the Genome Analysis Toolkit's Unified Genotyper[61]. These variants were filtered using VCFTOOLs[62] with the following specifications: Minor Allele Frequency of 0.05, Minimum Depth of 10x, and a Maximum Variant

Missing of 0.75. The resulting.vcf files provided a list of candidate bi-allelic polymorphisms to track at each time point, treatment, and phenotypic group in the larval samples. Accordingly, GATK's Haplotype Caller was used to identify these candidate polymorphisms within each larval alignment file, and the resulting.vcf was filtered using VCFTOOLS and the following specifications: Minor Allele Frequency of 0.01, Minimum Depth of 50x, and Maximum Depth of 450x. Only variants that passed quality filtering and were identified in all larval samples (i.e., each day, treatment, and phenotypic group) were retained for downstream analyses. This process resulted in a candidate SNP list of 29,400 variants. Allele frequencies for each variant were computed as the alternate allelic depth divided by total coverage at the locus.

**Allele frequency analysis**. To explore how the allele frequency of the 29,400 SNPs changed in each environment throughout the course of the experiment, we used a combination of PCA, outlier loci identification tests, and a statistical test of genomic differentiation ($F_{ST}$). We visualized patterns of genetic variation throughout the experiment with PCA (*prcomp* function in R). Prior to PCA, the allele frequency matrix was centered and scaled using the *scale* function in R. Only larval samples that encompassed the full phenotypic distribution within a particular bucket were included in this analysis. In other words, the rows of the allele frequency matrix corresponding to larval samples that were selectively segregated based on shell size were removed, and PCA was run using the day 0 larval population and larval samples collected from each treatment on days 6, 26, and 43. A separate PCA was then implemented using allele frequency data from the day 0 larval population and all day 6 larval samples, which included discrete size groups from each treatment. This analysis thus explicitly examined a genomic signature of the individuals that were phenotypically distinct.

We next sought to identify the presence, number, and treatment-level overlap of genetic variants that significantly changed in frequency between larval samples. Specifically, Fisher's Exact Test (FET) and the CMH test were used to generate probabilities of observed allele frequency changes, using the package *Popoolation*[63] in R. *P* values for each SNP were converted to *q* values in the R package *q value*[64], and significant SNPs were identified as those SNPs with a *q* value < 0.01. As the CMH test computes probabilities for SNPs based on consistent changes among replicates, it is a powerful approach to identifying significantly changed SNPs when treatment replicates are available[63]. Accordingly, this test was used identify significant allele frequency changes between the day 0 larval population and the day 26 ambient and low-pH treatment replicates ($N = 3$), the day 0 larval population and the settled individuals from ambient ($N = 2$) and low-pH treatment replicates ($N = 3$), and between the top 20% and bottom 80% of growers in each treatment ($N = 2$). This test thus produced a single, significant SNP list for each treatment on days 26 and 43, as well as the size-separated day 6 larvae. As treatment replicates were not available the ambient and low-pH larval population samples on day 6, an alternate contingency table test, FET, was used to obtain a list of significant SNPs for this day of sampling (with identical rank-based approach/ multiple testing corrections as those used for the CMH tests). The resulting lists of significant SNPs for days 6, 26, and 43 were then compared with identify outlier loci for each treatment. Specifically, outliers were identified for each treatment as those loci containing significant SNPs on each sampling day (i.e., SNPs overlapping among a treatment's three significant SNP lists). As the buckets sampled on day 6 were independent cultures from those buckets sampled on days 26 and 43, this process leverages both multiple independent larval cultures and sampling days to obtain a robust outlier list for each treatment.

To provide a third, independent metric of genomic change in the larval population throughout the experiment, we computed the $F_{ST}$ statistic for a series of comparisons. Specifically, we implemented a methods-of-moments estimator of $F_{ST}$ from Pool-seq data in an analysis of variance framework, as described in Hivert et al.[65] (*poolFstat* package in R). A global (exome-wide) $F_{ST}$ statistic was computed pairwise between the day 0 larval population and the day 6 ambient and low pH larvae replicate buckets, day 26 ambient and low-pH larvae replicate buckets, and settled individuals from all replicate buckets in ambient and low pH. $F_{ST}$ was also computed to compare differentiation between phenotypic groups (top 20% and bottom 80% of growers) on day 6. As *poolFstat* necessitates an approximation of population size, we parameterized the model using larval counts obtained on days 0 and 26, and estimates of larval population size for days 6 and 43. The population size in each replicate bucket on day 0 was 100,000 individuals, as larval counts were conducted on the starting larval culture and an equal volume (containing 100,000 individuals) was added to each replicate bucket. To calculate population sizes on day 26, 200 mL of seawater was extracted from each replicate bucket, from which larvae were concentrated using a 70 µm mesh filter and subsequently photographed using brightfield microscopy. This mesh size was selected based on data obtained from the size selection of larvae (Supplementary Fig. 1), and chosen to filter out D-veligers that had failed to grow, and therefore survive, beyond day 6. The number of photographed larvae on day 26 thus provided a proxy for the population size at this time point: population size was estimated to be 3685, 2090, and 1183 in the ambient replicates and 2503, 1733, and 2888 in the low-pH replicates. These estimates suggest average mortality rates of 97.7 and 97.6 % in the ambient and low-pH conditions, respectively. These estimates are in accordance with previous studies that have similarly reported substantial mortality of marine bivalve larvae

reared in laboratory settings[66–69]. Larval counts from day 26 were used to estimate a broad range of feasible population sizes on days 6 and 43. Specifically, given the observed population sizes on day 26 (reported above), $F_{ST}$ was computed over large ranges of population sizes on day 6 (10,000–40,000 individuals) to encompass both linear and nonlinear declines in population size throughout the growth period. While larval counts were additionally not conducted on day 43 (all larvae sampled on this day needed to be preserved for sequencing to ensure accurate allele frequency estimates), the larvae were observable to the naked eye at this time point. This allowed for the observations that there were: (1) no treatment-specific patterns in the total number of settled larvae and (2) the number of settled larvae was greater than 100 but far less than 400 individuals at this time point. $F_{ST}$ was thus computed using input pool sizes between 100 and 400 individuals for day 43 samples. Results demonstrating how computed values of $F_{ST}$ changed according to these differences in input pool size are reported in Supplementary Table 3.

**Gene identification/Ontologies**. We next sought to explore the biological pathways that were associated with survivorship in each pH treatment and/or size group during the experiment. To accomplish this, we indexed our list of outlier loci using the annotated transcriptome provided in Moreira et al.[23]. Their annotation utilizes NCBI's nucleotide and nonredundant, Swissprot, KEGG, and COG databases, thus providing a thorough survey of potential genes and pathways associated with our candidate SNPs. We generated gene lists for pH-specific outlier loci, which were identified as loci that showed signatures of selection on all sampling days and were unique to each environment. We also generated a candidate gene list for loci that exhibited shared signatures of selection in each treatment. These lists thus only contain robust outlier loci (i.e., containing significant SNPs in multiple independent replicates), with potentially strong effect sizes (i.e., containing significant SNPs at multiple developmental stages). Lastly, we used the Moreira et al.[23] annotation to explore the genes that exhibited signatures of selection for shell growth in ambient and low-pH conditions, as well as shared signatures of selection for shell size in each treatment.

**Reporting summary**. Further information on research design is available in the Nature Research Reporting Summary linked to this article.

## Data availability

Exome capture nucleotide sequences generated for this study are available as an NCBI SRA project (PRJNA578149). Data necessary to replicate all figures presented in the paper, as well as those figures and tables from the supplement are provided as a Source Data File. All raw data used in analyses are publicly available at: [https://github.com/MarkCBitter/Mgallo_SelectionExperiment].

## Code availability

Code used to analyze data and generate figures for this paper is publicly available at [https://github.com/MarkCBitter/Mgallo_SelectionExperiment].

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

## Acknowledgements

We thank Samir Alliouane for extensive technical assistance during the completion of the experiment. We would also like to thank Angelica Miglioli for experimental assistance, Régis Lasbleiz for microalgal supply for larval feeding, Jacob Enk at Arbor Biosciences for guidance during exome capture array design and generation of associated methods section, and the University of Chicago Genomics Core Facility for sequencing assistance. We thank D. Rice and T. Price for insightful comments on the paper. This research was supported by the National Science Foundation Graduate Research Fellowship Program under Grant No. 1746045 to M.C.B. and NSF OCE-1521597 to L.K. M.C.B. was supported by Department of Education Grant No. P200A150101. L.K. was also supported by the European Commission Horizon 2020 Marie Sklodowska-Curie Action (No. 747637). Research funding was provided by the France and University of Chicago Center FAACTs award to C.A.P. and M.C.B.

## Author contributions

M.C.B. conceived and designed the experiment with inputs from L.K., J.P.G., and C.A.P. M.C.B. and L.K. performed the experiment. M.C.B. completed molecular lab work, with exome capture and sequencing assistance from Arbor Biosciences and the University of Chicago Genomics Core. M.C.B. completed all bioinformatics, statistical, and computational analyses. M.C.B. wrote the paper with inputs from L.K., J.P.G., and C.A.P.

## Competing interests

The authors declare no competing interests.
