## [Peer Review File · Nature Communications]

Reviewers' comments:

Reviewer #1 (Remarks to the Author):

Bitter et al. Cryptic genetic variation

Bitter et al. present a very interesting study on laboratory selection of mussel larvae in two different pH/CO₂ environments and can find, for the first time, evidence for pH dependent selection of specific alleles from standing genetic variation. The experiment has been carried out well, with an impressive amount of families and a high overall technical standard. I find the study quite exciting but have points that should be addressed prior to publication.

Major points

1-'Cryptic genetic variation' and introduction section. I am not too fond of the term cryptic genetic variation. I would prefer to stick to 'standing genetic variation'. Whether the observed 'cryptic' alleles are really cryptic and not expressed in specific mussel reefs remains to be established. Generally, extensive standing genetic variation is maintained by multilocus balancing selection in populations of many marine animals with enormous effective population sizes living in variable habitats (where specific microhabitats / conditions can select for rare alleles). This has been discussed in a number of papers in the last five to ten years and I would suggest to discuss some of these in more detail (e.g. eel exemplified Gagnaire et al. 2012, sea urchin and coral work from Palumbi's lab – Bay & Palumbi 2014, Pespeni et al. 2013, copepods & abalone DeWit et al. 2014/2015, sea star wasting disease selection event Schiebelhut et al. 2018), rather than using model organism papers that were published prior to the sequencing revolution.

2-Larval mortality. It is puzzling to me why larval mortality was not determined. It is relatively easy to count the amount of larvae in a sampled volume of water from the treatment buckets to calculate density / mortality over time and it is an important parameter in selection experiments. Numbers of larvae / settled mussels are indeed given in lines 536-544, yet they are not discussed. Are these estimates or counts? If the latter, then the authors should give the exact numbers and variability between replicates, if the former I would like to know what the estimates are based on. I am also puzzled by the statement in lines 543-544. Please clarify. Based on the method section it appears to me that you should have data that allows for calculation of mortality. The data should be discussed in comparison to other studies using the same species. It is known that larval mortality in mytilids can be very high (>99% at times), and high mortality would help explain the strong shifts observed in figures 3 and 4.

3-Targets of selection. The discussion of the alleles enriched at low pH is quite short, yet this is the fascinating aspect of the study. I would recommend expanding it. For example, there is quite a lot information on the role of tyrosinases in shell and periostracum formation (check e.g. Hüning et al. 2013 Mar Biol, 2016 Mar Gen). I would compare your tyrosinase isoform to others that have been associated with periostracum / shell formation. I also would suggest to include a table / figure with SNP positions indicating whether they are amino acid changing or not. This could significantly enrich the discussion (further, one could team up with structural modelers that could analyze, whether a change in amino acid at a given position might influence protein properties, see e.g. Fields et al. 2015 J exp Biol for an excellent example how single aa substitutions can radically alter enzymatic properties, also note recent work from Somero and colleagues). Similarly, if qPCR / in situ data is already present I would strongly suggest to include it into this ms.

Specific points:

L56 Disagree, see comment 1.

L59 I would use / cite more recent studies with a marine focus.

L84 I would not use 'extreme'. There are mussel populations that encounter such conditions (see e.g. Thomsen et al. 2017 Sci Adv), and carbonate chemistry in boundary layers and in between mussel reefs with low water currents in some coastal systems may strongly differ from open ocean conditions.

L98 Very nice, excellent accomplishment.

L101 Keep in mind that hypoxic coastal systems such as your lagoon (and inorganic carbon enriched systems in general) will suffer from non-linear increases in pCO₂ in the future. So -0.4 pH is not necessarily outside of the range of values your model system may encounter in 100 years, see Melzner et al. 2013 Mar Biol for some calculations and discussion.

L103 As mentioned above, I would expect the presence of beneficial genetic variation that may be utilized in some habitats now already (which a thorough screening of adult mussels in different reefs might reveal).

L118 Exchange 'being' for 'becoming'

L119 Please add the size distribution for day 43 pelagic larvae and settled individuals.

L145 I don't understand. You should have data of volume of water sampled vs. larval densities for the size determinations? These should allow for calculation of mortality.

L168 High mortality in mussel larvae during the first few days is quite common.

L206 Quite interesting!

L224 I would not use 'recover' here, as mortality was likely very high (as in other studies using mussel larvae, see e.g. Thomsen et al. 2017 Sci Adv), so poor performers have likely left the experimental population by day 14.

L226 This is a very conservative statement considering that larval mortality is very high in most experiments published.

L252 Again, mortality data would significantly enrich this discussion.

L253 Grammar, sentence structure.

L261 I would like to have more information on the degree of starvation in your cultures: how much longer did the larval phase last in your experiment in comparison to published studies and what is the fraction of animals (in %) that reached the juvenile, benthic (attached stage) by the end of the experiment? This data should be present, as you scraped the settled individuals from your experimental vessel walls (line 400-404).

L269 You could discuss heritabilities for larval shell size here, see e.g. Sunday et al. (2011) Plos One, Thomsen et al. (2017) Sci Adv.

L288 Nice!

L290 Remove 'dramatic'

L302 In this context, maybe discuss Plough et al. (2016) Mol Ecol.

L313 Expand section, see above.

L337 This statement is much too strong for the data at hand.

L343 I think this statement is outdated considering the high number of papers in all model systems that have focused on such questions in the last decade (check literature on rapid adaptation, local adaptation).

L344 There is no such thing as environmental stasis.

L358 This sounds quite high to me, is this the standard temperature used for *M. galloprovincialis*?

L373 *M. edulis* larvae start feeding at day 2 at a similar thermal regime. What about *M. gallo*? I also do not understand, how feeding rate was adjusted in the flow-through seawater experimental tanks. Do I understand correctly that a pulse of algae was given, which was then diluted over time until the next feeding event? What is then the average algal cell concentration in the experimental tanks? How often were algal cell densities checked in the tanks?

L383 How was the 'subset' isolated? If a known volume was randomly sampled with a pipette from each tank, this should give you size distribution and density (mortality over time).

L389 Keep in mind that egg size and larval size can be family specific (see Sunday et al. 2011 Plos One) during early development in bivalves.

L538 'observed decline in culture density' – please explain.

L544 How?

Figures:

Fig. 1: change day 43 from N=3* to N=2-3

Fig. 2: I would like to see the day 43 size distribution here.

Supplement:

I could not find the tyrosinase sequence in your table. Also, the supplement should contain additional tables with sequences that form the basis for e.g. Fig. 6.

Papers to take a look at:

Bay & Palumbi 2014 Current Biology, [https://www.cell.com/current-biology/fulltext/S0960-9822\(14\)01353-0](https://www.cell.com/current-biology/fulltext/S0960-9822(14)01353-0)

DeWit et al. 2015 *Evolutionary Applications*, doi:10.1111/eva.12335
DeWit et al. 2014 *Nat Commun*, <https://www.nature.com/articles/ncomms4652>
Gagnaire et al 2012 *Genetics*, doi.10.1534/genetics.111.134825
Schiebelhut et al. 2018 *PNAS*, <https://www.pnas.org/content/115/27/7069>
Plough et al. 2016 *Mol Ecol*, <https://onlinelibrary.wiley.com/doi/abs/10.1111/mec.13524>

Reviewer #2 (Remarks to the Author):

Overall

This concisely written manuscript features an experiment designed to detect the genomic response to selection caused by increased ocean acidity (OA), and answer the question of whether there is standing genetic variation for OA resistance in natural populations. The experimental system is the Mediterranean blue mussel *Mytilus galloprovincialis*, and the experiment focuses on early life history phases. Embryos and larvae of this species have previously been shown to be both highly susceptible to fluctuations in OA, and there is phenotypic evidence from crosses that populations possess variation in the levels of susceptibility to OA. Thus *M. galloprovincialis* is excellent experimental choice for this question. I feel that the results will be of interest to a broad audience of evolutionary biologists and climate change scientists, but at the same time recommend revisions that recast the results in terms of the inferences that can be made about cryptic genetic variation, and to consider more stringent criteria for identifying outliers.

General comments

My first major comment has to do with the interpretation of the outlier results. Based on the set of unique outliers that appear in the high OA treatment, the authors conclude that cryptic genetic variation (CGV) and specifically genotype by environment interactions (G X E) are an explanation for what appears to be fairly strong response to selection due to high OA. Yet, with these data alone, I do not think one can discriminate among various CGV mechanisms (more below) or the alternative that non-CGV genetic variation for traits that reduce OA stress is segregating in populations of mussels and was artificially selected during this experiment.

To illustrate this point, it is first important to be clear on what CGV is, and what the potential mechanisms of CGV are. In terms of presenting the concept of CGV the introduction and discussion appear to be confusing two CGV mechanisms. If we use the definition of Paaby and Rockman's (2014), then: "The distinguishing feature of CGV is that the conditions that induce allelic effects are rare or absent in the history of the population, and this rarity limits the opportunities for selection to act on the variation and allows it to accumulate." Two classes of allelic effects which cause CGV are G X E interactions and genotype by genotype interactions (G X G). Further, the G X G may be due to both dominance and epistasis. The key to releasing this variation to natural selection is the ability to "induce" new variation. With G X E, trait values for each genotype change in the new environment so that the additive variance increases. In the case of epistasis, either novel allele frequencies among interacting loci result in novel multilocus genotypes which in turn increases additive variation. Lastly, with dominance a rare recessive allele becomes common enough in the new environment so that the phenotypic effect of the homozygous genotypes results in an increase in additive variation.

As written, it seems that G X E and G X G interactions are confused. In the introduction, the Authors explain G X E effects (starting on line 48) in the following way:

"have further shown that rapid adaptation via standing variation is oftentimes characterized by genotype-environment interactions, in which a particular genetic background is most fit in one environment, while an alternate genetic background leads to a fitness advantage when the environment shifts"

But G X E effects focus on a single locus or set of QTLs and how genotypic trait values change

different environments. In the discussion the Authors appear to make a similar error in defining G X E effects. On line 248 they write:

“These patterns display a classic genotype-environment interaction in which a particular genetic background exhibits a specific trait value in one environment (e.g. accelerated shell growth in ambient pH), while an alternate genetic background leads to the same trait value when the environment shifts.”

Contributing to the confusion is the use of the term “genetic background” above. This term is typically used to describe the epistatic effects of genomic variation on a locus (or set of QTLs) known to have major phenotypic effects, and not the other way around. If we think quantitatively, then the trait value of a QTL genotype depends on the allelic states of other loci, i.e. the genetic background.

If we agree on these definitions of how CGV works, then all three allelic effects (G X E, epistasis, and dominance) may potentially explain the set of candidate loci in the high OA treatment (Fig. 6b) and it seems to me there is no way of distinguishing among these three CGV mechanisms with the present data set. Without information on the individual genotypes and their connection to phenotype, I do not see how one can differentiate among the three possible mechanisms. Further, a hallmark of the presence of CGV is an increase in the phenotypic variance in the novel environment. But looking at the results in Fig. 2 the variance at each temporal sampling point look similar, with the caveat that selection is likely truncating the phenotypic variance by eliminating smaller individuals in high OA treatments from the earliest sampling points. Moreover, standing genetic variation that is not “cryptic” could also explain the outlier SNPs unique to the high OA treatment. We could imagine a situation in which variation in loci that maintain high growth rates in high OA could be maintained in populations, because larvae may occasionally be exposed to spatially or temporally varying OA gradients that are caused by high biological productivity or low salinity. The high OA treatment is simply driving natural selection on existing variation that facilitates high growth rates when pH drops.

My second major comment has to do with the criteria used to select outlier SNPs. To protect against false positives, the Authors use a Q-value approach, but I feel the paper could be strengthened if additional and consistent stringency were employed to identify outliers in the high OA treatment. In demographic applications of outlier analyses, a null model is constructed from the distribution of a population statistic from all the loci. In this application, allele frequencies are changing due to random patterns of mortality as well as the effects of natural selection so an appropriate null model is less clear. Fig. 3 clearly shows that SNP frequencies are changing in both the ambient and high OA cultures, and that not all high OA cultures are converging on the same allele frequencies (as evidenced by the variation in day 43 high OA cultures along PC axis 2). I suggest adding the criteria that all outliers with significant Q-values must be shared across developmental stages and replicates. To some extent, this criteria appears to have been used for the two annotated outliers brought up in the discussion, but does not generally appear to have been used for all the outliers presented in Fig. 4a and Fig. 6. From the schematic in Fig. 1, it appears there are replicates for size from Day 6 on. For sampling of allele frequencies there appear to be replicates from Days 26 and 43. As the Authors point out, it is interesting to know the developmental point when selection is the strongest, but from the point of view of avoiding excessive false positives in the set of outlier SNPs a shift in allele frequency should be apparent throughout the sampling points after a few days exposure to high OA.

Specific comments

Lines 48-50. See my comments above the definitions of G X E and G X G interactions.

Lines 57-60. For an example of CGV in a stress response to high OA in the blue mussel *Mytilus edulis* and *M. trossulus*, see Kingston et al. (2018).

Figure 3. Please connect re-sampled buckets with lines so we can see trajectory of evolution in multivariate space.

Line 142-143. Please explain what this means: "...and an associated increase in the influence of allele frequency "drift" among replicate buckets" Is it solely random mortality with respect to genotype or could there be individual selection going on specific to each bucket?

Lines 153-154. "...sampled on days 26 and 43 (Fig. 1), outlier SNPs observed on all three sampling days point to candidate loci that may be putatively under selection in each pH environment." I feel this is a robust approach to outlier detection, as well as using the criteria that outliers must occur in all three replicates, as above in the general comments.

Lines 188-194. Information on shared outliers among the slow and fast groups would be helpful, as they suggest that suggest common "growth" loci.

Lines 206-212. The use of the phrase "genetic background" is confusing here, since the outlier analysis is targeting candidate loci under selection. Genetic background is more commonly used to refer to the genotypic states of non-outlier loci.

Line 253. Typo/or grammatical error: "larvae reared in led to the observed"

Lines 253-271. This paragraph is speculative and could be edited out. I agree that there is likely selection due to the larval culture conditions (Fig. 3), but the causes of selection are difficult to identify without further experimentation.

Line 275. "While some of this treatment disparity may be an artifact (i.e. false positives in the low pH or false negatives in the ambient treatment), it is unlikely that this is the case for all the 277 unique SNPs identified." See my general comments on identifying outliers.

Line 283-286. These sentences need to be edited per my general comments.

Line 288. "genetic variation necessary to improve fitness in simulated ocean acidification" Typo.

Lines 294-296. "Our data not only suggest the role of cryptic genetic variation in rapid adaptation, but also demonstrate this phenomenon in the context of a non-model species subject to global change." See Kingston et al. (2018) or an example in blue mussels (*M. edulis* + *M. trossulus*) that uses a GWAS approach and phenotypes under climate stress.

Lines 313-316. "The low-pH specific loci we identified (loci with outlier SNPs in every replicate bucket and across all sampling days provide targets of natural selection as ocean acidification progresses." Agreed, but are these same criteria also being applied to the earlier analyses? See my general comments.

Lines 514-527. See my earlier comment on the criteria for identifying outliers.

Lines 520-521. "The FET was used to identify outliers between the day 0 larval population and the day 6 larval populations in each treatment (no treatment replicates were available for this comparison)." From Fig. 1 it looks like there are at least N=3 replicate buckets for all treatments and sampling days, including day 0. But from the text it seems that day "0" represents the start of the culture from the single population, and post-fertilization. Please clarify.

Lines 553-556. "We also generated a candidate gene list for loci that exhibited shared signatures of selection in each treatment. These lists thus only contain robust candidate loci (loci identified as outliers in multiple independent replicates), with potentially strong effect sizes (loci identified as outliers at multiple developmental stages)." These two criteria should be applied for all analyses.

References

Paaby, A. B. and Rockman, M. V. (2014). Cryptic genetic variation: evolution's hidden substrate. *Nature Reviews Genetics* 15, 247.

Kingston, S. E., Martino, P., Melendy, M., Reed, F. A. and Carlon, D. B. (2018). Linking genotype

to phenotype in a changing ocean: inferring the genomic architecture of a blue mussel stress response with genome-wide association. *Journal of Evolutionary Biology* 31, 346-361.

Signed
David B. Carlon

Authors' response to reviewers:

We thank the Reviewers for their time, positive comments, and constructive feedback on our manuscript. We have made the following major revisions:

1. We have altered the title, introduction, and discussion to more broadly focus on standing variation (as opposed to cryptic genetic variation) and its role in the results presented in this manuscript. We still highlight the potential role of cryptic genetic variation in driving the observed results in the Discussion section, though suggest the limitations of the present study to determine which mechanisms of CGV may be driving the results, including the relative contribution of CGV given the potential standing variation currently maintained within natural populations by spatially and temporally fluctuating selection pressures.
2. We have increased our stringency for the identification of “outlier loci” under selection in each pH environment.
3. We have expanded the discussion of candidate low pH outliers and provide preliminary qPCR data in this referee response.
4. We now report estimates of larval mortality within each pH treatment, including a new supplementary table demonstrating the robust nature our computed F_{ST} values to variation in input pool size.

In addition to the summary of major revisions described above, below we respond individually to each reviewer's comments.

Reviewer 1:

Bitter et al. present a very interesting study on laboratory selection of mussel larvae in two different pH/CO₂ environments and can find, for the first time, evidence for pH dependent selection of specific alleles from standing genetic variation. The experiment has been carried out well, with an impressive amount of families and a high overall technical standard. I find the study quite exciting but have points that should be addressed prior to publication.

#1 Reviewer 1- 'Cryptic genetic variation' and introduction section. I am not too fond of the term cryptic genetic variation. I would prefer to stick to 'standing genetic variation'. Whether the observed 'cryptic' alleles are really cryptic and not expressed in specific mussel reefs remains to be established. Generally, extensive standing genetic variation is maintained by multilocus balancing selection in populations of many marine animals with enormous effective population sizes living in variable habitats (where specific microhabitats / conditions can select for rare alleles). This has been discussed in a number of papers in the last five to ten years and I would suggest to discuss some of these in more detail (e.g. eel exemplified Gagnaire et al. 2012, sea urchin and coral work from Palumbi's lab – Bay & Palumbi 2014, Pespeni et al. 2013, copepods & abalone DeWit et al. 2014/2015, sea star wasting disease selection event Schiebelhut et al. 2018), rather than using model organism papers that were published prior to the sequencing revolution.

Response: We have edited the manuscript with this suggestion in mind. Specifically, we changed the MS title to “Standing variation fuels rapid adaptation to ocean acidification”, and

have removed the mention of cryptic genetic variation from the introduction to focus more broadly on the literature describing the role of standing variation in rapid evolutionary processes. We have additionally highlighted examples from marine systems, several of which were suggested by the reviewer, to describe the mechanisms maintaining variation in natural populations even amidst high levels of gene flow.

As our study does document processes that are characteristic of adaptation via cryptic variation, we find it important to suggest this process in the discussion. But, we now highlight the difficulties in distinguishing the extent to which the observed adaptive dynamics were indeed driven by standing variation that was cryptic, or simply variation currently maintained by fluctuating selection pressures.

#2 Reviewer 1 - Larval mortality. It is puzzling to me why larval mortality was not determined. It is relatively easy to count the amount of larvae in a sampled volume of water from the treatment buckets to calculate density / mortality over time and it is an important parameter in selection experiments. Numbers of larvae / settled mussels are indeed given in lines 536-544, yet they are not discussed. Are these estimates or counts? If the latter, than the authors should give the exact numbers and variability between replicates, if the former I would like to know what the estimates are based on. I am also puzzled by the statement in lines 543-544. Please clarify. Based on the method section it appears to me that you should have data that allows for calculation of mortality. The data should be discussed in comparison to other studies using the same species. It is known that larval mortality in mytilids can be very high (>99% at times), and high mortality would help explain the strong shifts observed in figures 3 and 4.

Response: We used direct larval counts to compute larval mortality on day 26. These counts were then used to make inferences of larval population size on days 6 and 43. Mortality data is now provided in the Results section (Line 127) and in the Methods (line 531-534).

The population sizes reported in the original MS (lines 543-544) were included in the context of computing the F_{ST} statistic (i.e. pool sizes are inputs to parameterize the *poolFStat* model). The input pool size on day 26 was generated from the mortality estimate now described on lines 531-534. We did not have larval counts on day 6 and 43, however, and the population sizes originally reported were estimates based on our day 26 larval counts. We have adjusted the language of the methods to (1) stress that days 0 and 26 were based on counts and days 6 and 43 were based on estimates and (2) highlight that the input pool size does, in fact, not change the results or conclusions drawn from patterns of F_{ST} . Below, we discuss in detail how our rough estimates of population size on days 6 and 43 were generated (abbreviated version of this is provided in MS lines 537-542).

On day 6, a known volume of subsampled culture water was taken from each replicate bucket to generate size distributions for each treatment. However, as cultures were still very dense, not all larvae concentrated within this subsample were photographed, as we only photographed enough larvae to generate an accurate estimate of the size distribution (~100 individuals per bucket). Therefore, we do not have a count of larvae in the subsamples from this sampling day, precluding us from a density estimate. We thus generated F_{ST} estimates for day 6 using a large range population sizes at this time point (10,000-90,000 between). The results from this simulation-based approach are presented in Supplementary Table S3, and clearly demonstrate the robust nature of the F_{ST} statistic when the input pool size is large. The input value reported in the original MS (25,000 individuals/treatment), was input pool size used in generating the values presented in Figure 3b. This value was chosen because it falls between the

known population sizes on Days 0 and 26 and reflects the likely non-linear reduction in population size throughout the larval pelagic phase. Specifically, previous work has shown heightened mortality in early relative to late D-veliger stages (Satuito et al. 1994 doi: 10.2331/fishsci.60.65; Fotel et al. 1999 doi:10.1016/S0022-0981(98)00136-1), and our previously published work has shown that the incidence of likely lethal morphological abnormalities occurs during the late-trochophore period (Kapsenberg et al. 2018 doi: 10.1098/rspb.2018.2381).

On day 43, we do not have photographs from subsampled larvae to conduct counts/density estimates, as all larvae sampled were preserved for sequencing (this was done to ensure accurate allele frequency estimates from the pooled samples). However, settled larvae were visible to the naked eye and from this we observed (1) no treatment-specific patterns in total number of settled larvae and (2) that the number of larvae were >100 but certainly <<400 in each bucket. From this, we once again computed estimates of F_{ST} across a range of feasible population sizes (100-400 individuals) to ensure that patterns of F_{ST} remained constant (Table S3).

#3 Reviewer 1 - Targets of selection. The discussion of the alleles enriched at low pH is quite short, yet this is the fascinating aspect of the study. I would recommend expanding it. For example, there is quite a lot information on the role of tyrosinases in shell and periostracum formation (check e.g. Hüning et al. 2013 Mar Biol, 2016 Mar Gen). I would compare your tyrosinase isoform to others that have been associated with periostracum / shell formation. I also would suggest to include a table / figure with SNP positions indicating whether they are amino acid changing or not. This could significantly enrich the discussion (further, one could team up with structural modelers that could analyze, whether a change in amino acid at a given position might influence protein properties, see e.g. Fields et al. 2015 J exp Biol for an excellent example how single aa substitutions can radically alter enzymatic properties, also note recent work from Somero and colleagues). Similarly, if qPCR / in situ data is already present I would strongly suggest to include it into this ms.

Response: We agree that this is a promising area of future research and have expanded the discussion of candidate genes (lines 291-302).

The qPCR and *in situ* hybridization data collection is still in progress. These data come from an independent experiment with a different experimental design (and collaborators). To avoid extensive additions to the methods section and delays associated with coordinating data collected by a different set of collaborators, including these data in the main text of the manuscript is beyond the scope of this study. However, we agree that these data solidify the conclusions drawn from this experiment. As such, we have included preliminary qPCR data for 3 genes of interest (Tyrosinase tyr3, HSP70, and Chitinase) below.

While exploration into the causal variants driving the observed dynamics, and the extent to which they may influence protein properties, is an intriguing and important direction of future research, this is outside the limits of the data collected for this manuscript. Specifically, exome capture (used in present study) enriches the protein-coding region of the genome, and does not enrich regulatory regions where the causal variants of complex traits (e.g. physiological adaptation to climate change) likely occur (see Albert and Kruglyak 2015, doi: <https://doi.org/10.1038/nrg3891>). In our dataset, many variants at outlier loci change dramatically in frequency, and therefore it would be misleading to suggest that those within

amino acid changing regions are potentially causal, when we have not captured/sequenced upstream regions where the true causal variants likely lie.

Figure S3 | Preliminary qPCR from candate genes described in main text. Normalized log-fold expression change (+/- standard deviation) of *M. galloprovincialis* larvae reared in ambient ($pH_T = 8.05$) and low pH ($pH_T = 7.4$) conditions and collected 28 hours post-fertilization (data from independent experiment).

Table S1 | Primer information for results displayed in Fig. S2. Target gene, primer design, and p. value (paired

Gene	Primer Design	p-Value
Tyrosinase tyr3	5'-CCCGAGCATGACATGGAGTT-3' 5'-GATCATGGGCAGCTGTCTCA-3'	0.01
HSP70	5'-GTAGTCGGTGGCCATTCAGA-3' 5'-GCAAGAGATGGCTGTTTCGG-3'	0.02*
Chitinase	5-CAAGGGACAGGGTACAGCAG-3' 5-GTCCAACCACCCTTCTGTCC-3'	0.005

T-test) associated with results presented in Fig. S2.

Reviewer 1 Specific comments:

#4 L56 Disagree, see comment 1.

Response: Resolved in MS. Also, see response to comment 1.

#5 L59 I would use / cite more recent studies with a marine focus.

Response: We have added more recent/marine focused studies including Schiebelhut et al. 2018, De Wit et al. 2014, Gagnaire et al. 2012, Silliman 2019, Bay and Palumbi 2014, Pespeni et al. 2013

#6 L84 I would not use ‘extreme’. There are mussel populations that encounter such conditions (see e.g. Thomsen *et al.* 2017 *Sci Adv*), and carbonate chemistry in boundary layers and in between mussel reefs with low water currents in some coastal systems may strongly differ from open ocean conditions.

Response: We recognize *Mytilus* populations inhabiting various coastal regions currently reside periodically experience pH values similar to the low pH treatment in this experiment -- e.g. coastal habitats in Northeast Pacific (Wootton *et al.* 2008 doi:10.1073/pnas.0810079105, Kwiatkowski *et al.* 2016 doi:10.1038/srep22984) and the Kiel Fjord in the Baltic Sea (Thomsen *et al.* 2017 doi: 10.1126/sciadv.1602411). In the discussion, we now highlight how the extensive gene flow among *Mytilus* populations may, in part, be driving the maintenance of low pH tolerant variation within the Mediterranean. However, our time-series data (SeaFET pH sensor) from a full year of sampling indicates that the low pH treatment condition ($\text{pH}_T=7.4$) falls far outside the variability experienced by mussels inhabiting the lagoon ($\sim\text{pH}_T$ 7.8-8.1; Kapsenberg *et al.* 2018 doi: 10.1098/rspb.2018.2381). For adults, or larvae produced by this specific population, we would thus consider 7.4 an “extreme” low pH condition.

#7 L98 Very nice, excellent accomplishment.

Response: Thank you!

#8 L101 Keep in mind that hypoxic coastal systems such as your lagoon (and inorganic carbon enriched systems in general) will suffer from non-linear increases in pCO_2 in the future. So -0.4 pH is not necessarily outside of the range of values your model system may encounter in 100 years, see Melzner *et al.* 2013 *Mar Biol* for some calculations and discussion.

Response: This is an insightful consideration and have altered lines 95-96 to now read: “Furthermore, while estimates of global mean seawater pH project a decline of -0.4 pH_T units by 2100¹⁰, marine species occupying unequilibrated coastal regions, such as the lagoon habitat of the study population, may periodically experience pH conditions that fall far below projected global means.” (Melzner *et al.* 2013; doi: 10.1007/s00227-012-1954-1).

#9 L103 As mentioned above, I would expect the presence of beneficial genetic variation that may be utilized in some habitats now already (which a thorough screening of adult mussels in different reefs might reveal).

Response: We agree and have added information to reflect this point in the introduction lines 49-52 and discussion lines 251-259.

#10 L118 Exchange ‘being’ for ‘becoming’

Response: Resolved in MS.

#11 L119 Please add the size distribution for day 43 pelagic larvae and settled individuals.

Response: This is not possible as all individuals collected on day 43 were used for sequencing. Specifically, this was the sampling point in which the population sizes were the smallest. Accurate poolseq allele frequency estimates hinge upon maximizing pool size (see Fracassetti *et al.* 2015; doi: 10.1371/journal.pone.0140462), so all individuals collected at this point were reserved for sequencing. The larval population size on day 43 had been reduced to well below 1000 individuals (as stated in reply to comment 2, settled individuals were

observable by the naked eye) and as any individuals that are used for shell size estimates are lost from the sequencing pool, we chose to not phenotype individuals on this sampling day.

#12 L145 I don't understand. You should have data of volume of water sampled vs. larval densities for the size determinations? These should allow for calculation of mortality.

Response: We have added a mortality estimate between days 0 and 26 for each treatment. Please refer to Results lines 127 and Methods Lines 531-534, as well as the response to the major point raised above. Briefly, to obtain larvae for size determinations in our experiment, we extracted 200 mL of seawater from each replicate tanks, concentrated larvae in a mesh filter, transferred to a centrifuge tube, and ~1 mL aliquots from the centrifuge tube were added to a glass slide for photography until enough larvae to generate size distributions were photographed. This provides the density as long as all larvae isolated from the original 200 mL were photographed. This was indeed done on day 26 (again, see response to the major point. However, cultures are so dense on day 6 that not all larvae isolated in the subset volume needed to be photographed, so we unfortunately do not have counts of larvae from the replicates on this sampling day. Additionally, sizing did not occur at all on day 43 to preserve all larvae collected for sequencing (see response to minor comment above).

#13 L168 High mortality in mussel larvae during the first few days is quite common.

Response: We have noted this and added references identifying similar mortality rates to those we estimated in lines 534-536.

#14 L206 Quite interesting!

#15 L224 I would not use 'recover' here, as mortality was likely very high (as in other studies using mussel larvae, see e.g. Thomsen et al. 2017 Sci Adv), so poor performers have likely left the experimental population by day 14.

Response: This sentence has been changed to read: "...after which the interplay of natural selection and acclimation drove a partial convergence of the size distributions in the ambient and low pH treatments."

#16 L226 This is a very conservative statement considering that larval mortality is very high in most experiments published.

Response: We have adjusted this sentence as follows: "It is likely that partial convergence of shell size distributions observed by day 14 was, at least in part, driven by natural selection on pH tolerance." It is important to note that while mass mortality was indeed observed in both treatments, this does not allow inference regarding pH-specific signatures of selection. Therefore, this paragraph (lines 187-195) describes distinguishing dynamics of the phenotypic distributions between environments that indicate alternate selective regimes.

#17 L252 Again, mortality data would significantly enrich this discussion.

Response: As noted above, mortality data has been added to the MS.

#18 L253 Grammar, sentence structure.

Response: Resolved in MS.

#19 L261 *I would like to have more information on the degree of starvation in your cultures: how much longer did the larval phase last in your experiment in comparison to published studies and what is the fraction of animals (in %) that reached the juvenile, benthic (attached stage) by the end of the experiment? This data should be present, as you scraped the settled individuals from your experimental vessel walls (line 400-404).*

Response: It is now explicitly stated in the MS how the observed time to settlement compares to published studies. We have added estimates of numbers of individuals settled on day 43.

#20 L269 *You could discuss heritabilities for larval shell size here, see e.g. Sunday et al. (2011) Plos One, Thomsen et al. (2017) Sci Adv.*

Response: A sentence calling attention to the suggested publication documenting the heritability of low pH sensitivity is now in line 231.

#21 L288 Nice!

#22 L290 Remove 'dramatic'

Response: Sentence no longer in revised MS.

#23 L302 *In this context, maybe discuss Plough et al. (2016) Mol Ecol.*

Response: This reference has been added to line 316.

#24 L313 *Expand section, see above.*

Response: Resolved (see response to comment above).

#25 L337 *This statement is much too strong for the data at hand.*

Response: Sentence changed to – "...harbors standing variation that would facilitate adaptation to ocean acidification..." (line 325-326).

#26 L343 *I think this statement is outdated considering the high number of papers in all model systems that have focused on such questions in the last decade (check literature on rapid adaptation, local adaptation).*

Response: Sentence changed to read: "...the substantial levels of genetic variation present in natural populations *historically* puzzled evolutionary biologists."

#27 L344 *There is no such thing as environmental stasis.*

Response: "environmental stasis" removed from MS.

#28 L358 *This sounds quite high to me, is this the standard temperature used for *M. galloprovincialis*?*

Response: The temperature on L358 in the previous version of the MS is in the context of spawning the adult mussels, in which we thermally stressed the individuals to facilitate the release of gametes. This does not refer to the experimental conditions under which the larvae were reared, which was indeed based on our time-series temperature data collected within the

study population's natural habitat during spawning season (17.2°C; see Table S2).

#29 L373 M. edulis larvae start feeding at day 2 at a similar thermal regime. What about M. gallo? I also do not understand, how feeding rate was adjusted in the flow-through seawater experimental tanks. Do I understand correctly that a pulse of algae was given, which was then diluted over time until the next feeding event? What is then the average algal cell concentration in the experimental tanks? How often were algal cell densities checked in the tanks?

Response: During extensive pilot/previous experiments using this population of *M. gallo* (see Kapsenberg *et al.* 2018 doi: 10.1098/rspb.2018.2381), we did not observe larvae assimilating food into the gut until after day 3. As this experiment ran for a substantial duration, we thus wanted to limit the risk of biofouling in the experimental system, and thus did not begin feeding on day 4 when we were certain the larvae would utilize the added algae.

The interpretation of the feeding scheme is correct – algae were added as a pulse to each experimental replicate, twice daily (early morning, late afternoon). As the system was flow through on days 0-26, the density of algae declined between consecutive pulses. The number of algae in each pulse was determined daily by conducting counts of the algal stock solution. The reported value is the average number of algae added to the replicates each day. Ultimately, larval growth/settlement indicates larvae were feeding, while the decline in algal density between feeding is further evidence that the larvae were food limited.

#30 L383 How was the 'subset' isolated? If a known volume was randomly sampled with a pipette from each tank, this should give you size distribution and density (mortality over time).

Response: As described above, the MS now clarifies this on line 529. 200 mL of seawater was extracted from the replicate tanks, larvae were concentrated in a mesh filter, transferred to a centrifuge tube, and ~1 mL aliquots from the centrifuge tube were added to a glass slide for photography until enough larvae to generate size distributions were photographed. This provides the density as long as all larvae isolated were photographed. This was indeed done on day 26 (again, see response to the major point). However, cultures are so dense on day 6 that not all larvae isolated in the subset volume of 200 mL needed to be photographed, as we only photograph enough larvae to generate size distribution estimates (~100 per replicate), and far more than 100 larvae were contained in the subset.

#31 L389 Keep in mind that egg size and larval size can be family specific (see Sunday et al. 2011 Plos One) during early development in bivalves.

Response: Yes, we have evidence of this from pilot experiments, though did not find it pertinent within the present MS, as we did not trace the surviving genotypes to their founding parent/haplotype.

#32 L538 'observed decline in culture density' – please explain.

Response: Addressed in responses above (see lines 531-534).

#33 L544 How?

Response: See comments above. Mortality estimate now provided in Results line 127 and Methods line 531-534.

Figures:

#34 *Fig. 1: change day 43 from N=3* to N=2-3*

Response: This has been resolved.

#35 *Fig. 2: I would like to see the day 43 size distribution here.*

Response: Unfortunately, all individuals sampled on day 43 were preserved for sequencing and we cannot generate size estimates (see response to minor comment above).

Supplement:

#36 *I could not find the tyrosinase sequence in your table. Also, the supplement should contain additional tables with sequences that form the basis for e.g. Fig. 6.*

Response: The accession codes for sequences our outliers aligned to are now provided in the main text (lines 286, 293, and 294). While the original supplemental file contains the annotations of outliers, we have now added a series of supplemental .fasta files containing all outliers and their sequences (Supplementary files 2-4 for outlier loci sequences, Supplementary files 6-8 for shell growth genes).

Reviewer 2:

*This concisely written manuscript features an experiment designed to detect the genomic response to selection caused by increased ocean acidity (OA), and answer the question of whether there is standing genetic variation for OA resistance in natural populations. The experimental system is the Mediterranean blue mussel *Mytilus galloprovincialis*, and the experiment focuses on early life history phases. Embryos and larvae of this species have previously been shown to be both highly susceptible to fluctuations in OA, and there is phenotypic evidence from crosses that populations possess variation in the levels of susceptibility to OA. Thus *M. galloprovincialis* is excellent experimental choice for this question. I feel that the results will be of interest to a broad audience of evolutionary biologists and climate change scientists, but at the same time recommend revisions that recast the results in terms of the inferences that can be made about cryptic genetic variation, and to consider more stringent criteria for identifying outliers.*

#37 *Reviewer 2: My first major comment has to do with the interpretation of the outlier results. Based on the set of unique outliers that appear in the high OA treatment, the authors conclude that cryptic genetic variation (CGV) and specifically genotype by environment interactions (G X E) are an explanation for what appears to be fairly strong response to selection due to high OA. Yet, with these data alone, I do not think one can discriminate among various CGV mechanisms (more below) or the alternative that non-CGV genetic variation for traits that reduce OA stress is segregating in populations of mussels and was artificially selected during this experiment.*

To illustrate this point, it is first important to be clear on what CGV is, and what the potential mechanisms of CGV are. In terms of presenting the concept of CGV the introduction and discussion appear to be confusing two CGV mechanisms. If we use the definition of Paaby and Rockman's (2014), then:

“The distinguishing feature of CGV is that the conditions that induce allelic effects are rare or absent in the history of the population, and this rarity limits the opportunities for selection to act on the variation and allows it to accumulate.” Two classes of allelic effects which cause CGV are G X E interactions and genotype by genotype interactions (G X G). Further, the G X G may be due to both dominance and epistasis. The key to releasing this variation to natural selection is the ability to “induce” new variation.

With G X E, trait values for each genotype change in the new environment so that the additive variance increases. In the case of epistasis, either novel allele frequencies among interacting loci result in novel multilocus genotypes which in turn increases additive variation. Lastly, with dominance a rare recessive allele becomes common enough in the new environment so that the phenotypic effect of the homozygous genotypes results in an increase in additive variation.

As written, it seems that G X E and G X G interactions are confused. In the introduction, the Authors explain G X E effects (starting on line 48) in the following way:

“have further shown that rapid adaptation via standing variation is oftentimes characterized by genotype-environment interactions, in which a particular genetic background is most fit in one environment, while an alternate genetic background leads to a fitness advantage when the environment shifts”

But G X E effects focus on a single locus or set of QTLs and how genotypic trait values change different environments. In the discussion the Authors appear to make a similar error in defining G X E effects. On line 248 they write:

“These patterns display a classic genotype-environment interaction in which a particular genetic background exhibits a specific trait value in one environment (e.g. accelerated shell growth in ambient pH), while an alternate genetic background leads to the same trait value when the environment shifts.”

Contributing to the confusion is the use of the term “genetic background” above. This term is typically used to describe the epistatic effects of genomic variation on a locus (or set of QTLs) known to have major phenotypic effects, and not the other way around. If we think quantitatively, then the trait value of a QTL genotype depends on the allelic states of other loci, i.e. the genetic background.

If we agree on these definitions of how CGV works, then all three allelic effects (G X E, epistasis, and dominance) may potentially explain the set of candidate loci in the high OA treatment (Fig. 6b) and it seems to me there is no way of distinguishing among these three CGV mechanisms with the present data set. Without information on the individual genotypes and their connection to phenotype, I do not see how one can differentiate among the three possible mechanisms. Further, a hallmark of the presence of CGV is an increase in the phenotypic variance in the novel environment. But looking at the results in Fig. 2 the variance at each temporal sampling

point look similar, with the caveat that selection is likely truncating the phenotypic variance by eliminating smaller individuals in high OA treatments from the earliest sampling points. Moreover, standing genetic variation that is not “cryptic” could also explain the outlier SNPs unique to the high OA treatment. We could imagine a situation in which variation in loci that maintain high growth rates in high OA could be maintained in populations, because larvae may occasionally be exposed to spatially or temporally varying OA gradients that are caused by high biological productivity or low salinity. The high OA treatment is simply driving natural selection on existing variation that facilitates high growth rates when pH drops.

Response: We are appreciative for these thoughtful comments and discussion regarding distinguishing if/how cryptic variation may have contributed to the observed results, and have adjusted the manuscript accordingly. We have changed the title to “Standing variation fuels rapid adaptation to ocean acidification.” We have altered the introduction to focus more broadly on standing variation and its role in rapid adaptation, rather than extensively describing CGV and its potential role in adaptation to climate change. We have added data in the Results section demonstrating that phenotypic variation (as determined by coefficient of variation in larval size) is indeed elevated in the low pH treatment on Days 3 and 7 (this was difficult to observe in the scaled Figure 2 of larval shell size patterns). In the Discussion, we still highlight the potential role that CGV may have played in the observed dynamics. Additionally, we explicitly state the limitations of the present data to determine the relative contribution of standing variation (maintained in natural populations via balanced polymorphisms) vs. cryptic variation in driving the observed results. Lastly, we address the complication that CGV mechanisms other than G X E interactions may be driving the observed results. We also highlight a suggested reference (Kingston et al. 2018) as demonstration of the release of hidden variation in stressful environments in *Mytilus* mussels.

#38 Reviewer 2: My second major comment has to do with the criteria used to select outlier SNPs. To protect against false positives, the Authors use a Q-value approach, but I feel the paper could be strengthened if additional and consistent stringency were employed to identify outliers in the high OA treatment. In demographic applications of outlier analyses, a null model is constructed from the distribution of a population statistic from all the loci. In this application, allele frequencies are changing due to random patterns of mortality as well as the effects of natural selection so an appropriate null model is less clear. Fig. 3 clearly shows that SNP frequencies are changing in both the ambient and high OA cultures, and that not all high OA cultures are converging on the same allele frequencies (as evidenced by the variation in day 43 high OA cultures along PC axis 2). I suggest adding the criteria that all outliers with significant Q-values must be shared across developmental stages and replicates. To some extent, this criteria appears to have been used for the two annotated outliers brought up in the discussion, but does not generally appear to have been used for all the outliers presented in Fig. 4a and Fig. 6. From the schematic in Fig. 1, it appears there are replicates for size from Day 6 on. For sampling of allele frequencies there appear to be replicates from Days 26 and 43. As the Authors point out, it is interesting to know the developmental point when selection is the strongest, but from the point of view of avoiding excessive false positives in the set of outlier SNPs a shift in allele frequency should be apparent throughout the sampling points after a few days exposure to high OA.

Response: We agree that a more stringent approach for identifying outliers would strengthen the manuscript and have adjusted our criteria. We now explicitly distinguish between “significant SNPs” and “outlier loci.” Significant SNPs (previously “Outlier SNPs”) are those

SNPs passing our Q-value threshold. “Outlier loci” now refers to the candidate pH gene list we previously included in Figure 6 (loci containing a “significant SNP” that was significant across all sampling days/developmental points within a treatment). As suggested, this stringency dramatically reduces potential false positives, and we now effectively leverage our multiple independent larval cultures and sampling days. It should be noted that this new criterion changes the results presented in Figure 4a, as SNPs identified as “Outlier SNPs” in the first version of the MS are now considered “Significant SNPs”. We have moved this figure to the supplement (Fig. S1) and changed the y-axis to read “Significant SNPs”.

Reviewer 2 Specific comments:

#39 Lines 48-50. See my comments above the definitions of G X E and G X G interactions.

Response: Resolved. See response to major revision above.

*#40 Lines 57-60. For an example of CGV in a stress response to high OA in the blue mussel *Mytilus edulis* and *M. trossulus*, see Kingston et al. (2018).*

Response: This example of CGV has been added to the Discussion on lines 273-276. Thank you for the suggestion!

#41 Figure 3. Please connect re-sampled buckets with lines so we can see trajectory of evolution in multivariate space.

Response: This has been resolved.

#42 Line 142-143. Please explain what this means: “...and an associated increase in the influence of allele frequency “drift” among replicate buckets” Is it solely random mortality with respect to genotype or could there be individual selection going on specific to each bucket?

Response: We have clarified this in the manuscript lines 124-125. We do presume both suggested mechanisms (random mortality and bucket-effects) led to this increase in variation among replicates. The MS now reads – “...and an associated increase in the influence of allele frequency “drift” among replicate buckets (i.e. random mortality and leading to replicate-specific patterns of selection at loci unresponsive to pH treatment)”.

#43 Lines 153-154. “...sampled on days 26 and 43 (Fig. 1), outlier SNPs observed on all three sampling days point to candidate loci that may be putatively under selection in each pH environment.” I feel this is a robust approach to outlier detection, as well as using the criteria that outliers must occur in all three replicates, as above in the general comments.

Response: We agree, please see response to the major point above, and refer to the MS, lines 136-137.

#44 Lines 188-194. Information on shared outliers among the slow and fast groups would be helpful, as they suggest that suggest common “growth” loci.

Response: These shared growth loci are contained in a tab of Supplementary File 5. Associated sequences for these outliers found in Supplementary File 6-8.

We feel a discussion of these loci is beyond the scope of this study, which aims to demonstrate and describe low pH-specific patterns of selection.

#45 Lines 206-212. *The use of the phrase “genetic background” is confusing here, since the outlier analysis is targeting candidate loci under selection. Genetic background is more commonly used to refer to the genotypic states of non-outlier loci.*

Response: We have adjusted this to read: “Therefore, 76% of loci associated with fast shell growth in low pH were not associated with fast growth in the ambient treatment, indicating unique targets of selection on shell growth in each environment”

#46 Line 253. *Typo/or grammatical error: “larvae reared in led to the observed”*

Response: Resolved in MS.

#47 Lines 253-271. *This paragraph is speculative and could be edited out. I agree that there is likely selection due to the larval culture conditions (Fig. 3), but the causes of selection are difficult to identify without further experimentation.*

Response: While we our experimental design does not specifically probe the extent to which food-augmented acclimation may have driven the convergence of the size distributions, we feel this is important to include in the MS as there is a substantial literature demonstrating the role of food availability on the observed effects of OA conditions (see references: Pansch *et al.* 2014 doi: 10.1111/gcb.12478; Thomsen *et al.* 2013 doi: 10.1111/gcb.12109)

#48 Line 275. *“While some of this treatment disparity may be an artifact (i.e. false positives in the low pH or false negatives in the ambient treatment), it is unlikely that this is the case for all the 277 unique SNPs identified.” See my general comments on identifying outliers.*

Resolved: See response to major comments above and manuscript lines 136-137.

#49 Line 283-286. *These sentences need to be edited per my general comments.*

Response: Resolved, see response to major revision above.

#50 Line 288. *“genetic variation necessary to improvs fitness in simulated ocean acidification” Typo.*

Response: Resolved in MS

#51 Lines 294-296. *“Our data not only suggest the role of cryptic genetic variation in rapid adaptation, but also demonstrate this phenomenon in the context of a non-model species subject to global change.” See Kingston *et al.* (2018) or an example in blue mussels (*M. edulis* + *M. trossulus*) that uses a GWAS approach and phenotypes under climate stress.*

Response: This has been referenced/described in lines 273-276.

#52 Lines 313-316. *“The low-pH specific loci we identified (loci with outlier SNPs in every replicate bucket and across all sampling days provide targets of natural selection as ocean acidification progresses.” Agreed, but are these same criteria also being applied to the earlier analyses? See my general comments.*

Response: This comment has been addressed in the general comment above.

#53 Lines 514-527. *See my earlier comment on the criteria for identifying outliers.*

Response: Resolved, see Lines 136-137.

#54 Lines 520-521. “The FET was used to identify outliers between the day 0 larval population and the day 6 larval populations in each treatment (no treatment replicates were available for this comparison).” From Fig. 1 it looks like there are at least N=3 replicate buckets for all treatments and sampling days, including day 0. But from the text it seems that day “0” represents the start of the culture from the single population, and post-fertilization. Please clarify.

Response: We have clarified the text to indicate that the allele frequency sample generated from the Day 0 is not replicated (lines 512-515). We note that Figure 1 explicitly indicates the amount of replication for all “Allele Frequency” samples. For example, below the Day 0 portion of the schematic, the figure reads: “ **Allele Freq.** N = 1).

#55 Lines 553-556. “We also generated a candidate gene list for loci that exhibited shared signatures of selection in each treatment. These lists thus only contain robust candidate loci (loci identified as outliers in multiple independent replicates), with potentially strong effect sizes (loci identified as outliers at multiple developmental stages).” These two criteria should be applied for all analyses.

Response: Resolved, see above. This criterion is now applied for our lists of outlier loci.

References

Paaby, A. B. and Rockman, M. V. (2014). Cryptic genetic variation: evolution’s hidden substrate. Nature Reviews Genetics 15, 247.

Kingston, S. E., Martino, P., Melendy, M., Reed, F. A. and Carlon, D. B. (2018). Linking genotype to phenotype in a changing ocean: inferring the genomic architecture of a blue mussel stress response with genome-wide association. Journal of Evolutionary Biology 31, 346-361.

Signed

David B. Carlon

REVIEWERS' COMMENTS:

Reviewer #1 (Remarks to the Author):

The ms has been carefully revised and my criticism has been addressed. I have a few minor comments re description of methods / results. Space is not limiting I believe, so please make sure to describe methods / results as carefully as possible.

Line 359 please expand. Describe that food concentration declined due to dilution with seawater and due to consumption by larvae and that the food algae concentrations were not assessed over time. Essentially information provided in response to comment #29.

Line 530 please provide individual values for all 3 replicate culture buckets.

Line 534 disagree. Please give mesh width. Typically, no or very few larvae should be lost if one carefully washes nets.

Line 537 expand. It would be important to give the full rationale for your sensitivity analysis presented in Table S3 here. Essentially the information provided in the response letter.

Reviewer #2 (Remarks to the Author):

Review of "Standing genetic variation fuels rapid adaptation to ocean acidification"
Resubmission to Nature Communications

M. C. Bitter*, L. Kapsenberg, J. P. Gattuso, and C. A. Pfister

This is a greatly improved re-submission, and I appreciate the Author's careful consideration of my comments on the original submission. The focus on standing genetic variation greatly improves the flow of the manuscript, and the discussion of cryptic genetic variation in the discussion is now appropriate.

I have three recommendations before publication.

1. Outliers

The criteria for outlier loci remain unclear in the text, and are only presented in the Results section as far as I can tell. In the Results section the Authors write:

L136 "significant SNPs observed on all three sampling days point to candidate loci, hereafter termed outlier loci, that may be putatively under selection in each pH environment."

But I (still) would like to know the criteria are for outlier loci.

In their response to my original comment (#38), the Authors write this in the rebuttal letter:

"Response: We agree that a more stringent approach for identifying outliers would strengthen the manuscript and have adjusted our criteria. We now explicitly distinguish between "significant SNPs" and "outlier loci." Significant SNPs (previously "Outlier SNPs") are those SNPs passing our Q-value threshold. "Outlier loci" now refers to the candidate pH gene list we previously included in Figure 6 (loci containing a "significant SNP" that was significant across all sampling days/developmental points within a treatment). As suggested, this stringency dramatically reduces potential false positives, and we now effectively leverage our multiple independent larval cultures and sampling days. It should be noted that this new criterion changes the results presented in Figure 4a, as SNPs identified as "Outlier SNPs" in the first version of the MS are now considered "Significant SNPs". We have moved this figure to the supplement (Fig. S1) and changed the y-axis to read "Significant SNPs"."

But I was suggesting that outlier loci should be significant in all replicate cultures for all sampling days. In the response above, the Authors seem to be implying this, if they are indeed: "effectively

leverage our multiple independent larval cultures and sampling days." Is an outlier locus significant for each replicate within sampling days, or just one? Similarly, for the slow and fast growers, it appears that FET tests were done on the pooled data from the two replicates, rather than independent tests on each replicate. This might have been necessary if sample sizes ended up small. Please add the appropriate language to describe how outliers were defined. I suggest the best place for this content is at the end of the paragraph starting on L500 "We sought to identify..." and some appropriate leading text in the Results section.

The precise nature of the outlier criteria is important because the paper is pointing the research community towards candidate loci that ultimately requires experimental validation to show they are linked to physiological processes that reduce pH stress or aid growth in stressful environments. False positives are a pervasive problem in population genomics, and the construction and validation of appropriate null models to limit false positives can potentially increase the efficiency of the validation step.

2. L142 "Therefore, 58% of the loci exhibiting signatures of selection in the low pH treatment were unresponsive, and putatively neutral, in the ambient treatment (Fig. 3c), highlighting the polygenic nature of low pH adaptation and indicating that natural populations currently harbor variation at these putatively adaptive loci."

Maybe I'm missing something but I am not sure how the 58% was calculated. Is this sentence referring to the fraction of total outliers that are unique to the low pH treatment ($25/88 = 0.318$)? Perhaps the Authors are referring to significant vs. outlier loci?

3. L231 "As pH tolerance has been shown to exhibit heritability in *Mytilus* spp.³⁵ Kingston et al. 2018 (cited in the bibliography as #40) have also shown significant heritability for calcification rates under OA stress.

Sincerely,
David B. Carlon

Authors' response to Reviewers:

We thank the Reviewers for their time and constructive feedback throughout the revisions of this manuscript. We address the final minor comments below.

Note: Line numbers referenced in this response correspond to line numbers in the MS text file in which changes are tracked (Bitter_etal_2019_Final_TrackChanges.docx).

Reviewer 1:

The ms has been carefully revised and my criticism has been addressed. I have a few minor comments re description of methods / results. Space is not limiting I believe, so please make sure to describe methods / results as carefully as possible.

#1 Reviewer 1 - Line 359 please expand. Describe that food concentration declined due to dilution with seawater and due to consumption by larvae and that the food algae concentrations were not assessed over time. Essentially information provided in response to comment #29.

Response: This has been resolved in text on lines 471-477. As suggested, the detailed information provided in first round of referee responses has been added to the manuscript.

#2 Reviewer 1 - Line 530 please provide individual values for all 3 replicate culture buckets.

Response: This has been resolved in the updated manuscript on Lines 689-690.

#3 Reviewer 1 - Line 534 disagree. Please give mesh width. Typically, no or very few larvae should be lost if one carefully washes nets.

Response: We have included mesh size and removed the suggestion that we may have overestimated mortality. We highlight that our estimates are indeed in accordance with previous estimates for the species (Lines 685-687).

#4 Reviewer 1 - Line 537 expand. It would be important to give the full rationale for your sensitivity analysis presented in Table S3 here. Essentially the information provided in the response letter.

Response: As suggested, we have expanded upon this by including the detailed information provided in the original referee response in the updated manuscript. These additions are found on Lines 694-720.

Reviewer #2

This is a greatly improved re-submission, and I appreciate the Author's careful consideration of my comments on the original submission. The focus on standing genetic variation greatly improves the flow of the manuscript, and the discussion of cryptic genetic variation in the discussion is now appropriate.

#1 Reviewer 2 - *The criteria for outlier loci remain unclear in the text, and are only presented in the Results section as far as I can tell. In the Results section the Authors write:*

L136 "significant SNPs observed on all three sampling days point to candidate loci, hereafter termed outlier loci, that may be putatively under selection in each pH environment."

But I (still) would like to know the criteria are for outlier loci.

In their response to my original comment (#38), the Authors write this in the rebuttal letter: “Response: We agree that a more stringent approach for identifying outliers would strengthen the manuscript and have adjusted our criteria. We now explicitly distinguish between “significant SNPs” and “outlier loci.” Significant SNPs (previously “Outlier SNPs”) are those SNPs passing our Q-value threshold. “Outlier loci” now refers to the candidate pH gene list we previously included in Figure 6 (loci containing a “significant SNP” that was significant across all sampling days/developmental points within a treatment). As suggested, this stringency dramatically reduces potential false positives, and we now effectively leverage our multiple independent larval cultures and sampling days. It should be noted that this new criterion changes the results presented in Figure 4a, as SNPs identified as “Outlier SNPs” in the first version of the MS are now considered “Significant SNPs”. We have moved this figure to the supplement (Fig. S1) and changed the y-axis to read “Significant SNPs”.”

But I was suggesting that outlier loci should be significant in all replicate cultures for all sampling days. In the response above, the Authors seem to be implying this, if they are indeed: “effectively leverage our multiple independent larval cultures and sampling days.” Is an outlier locus significant for each replicate within sampling days, or just one? Similarly, for the slow and fast growers, it appears that FET tests were done on the pooled data from the two replicates, rather than independent tests on each replicate. This might have been necessary if sample sizes ended up small. Please add the appropriate language to describe how outliers were defined. I suggest the best place for this content is at the end of the paragraph starting on L500 “We sought to identify...” and some appropriate leading text in the Results section.

The precise nature of the outlier criteria is important because the paper is pointing the research community towards candidate loci that ultimately requires experimental validation to show they are linked to physiological processes that reduce pH stress or aid growth in stressful environments. False positives are a pervasive problem in population genomics, and the construction and validation of appropriate null models to limit false positives can potentially increase the efficiency of the validation step.

Response: We agree that clearly defining the criterion used for outlier loci identification is critical in such studies. Accordingly, we have altered the manuscript as suggested to more clearly define our criterion in lines 183-185 (Results) and lines 610-627 (Methods). We also highlight that we have already begun linking many of the outlier loci identified (including HSP70, Chitinase, Tyrosinase, Ubiquitin Protein Ligase, Kinesin) to physiological processes in response to low pH conditions using a combination of qPCR and in situ hybridizations (Kapsenberg, Bitter *et al.* in prep).

#2 Reviewer 2 - L142 “Therefore, 58% of the loci exhibiting signatures of selection in the low pH treatment were unresponsive, and putatively neutral, in the ambient treatment (Fig. 3c), highlighting the polygenic nature of low pH adaptation and indicating that natural populations currently harbor variation at these putatively adaptive loci.”

Maybe I’m missing something but I am not sure how the 58% was calculated. Is this sentence referring to the fraction of total outliers that are unique to the low pH treatment (25/88 = 0.318)? Perhaps the Authors are referring to significant vs. outlier loci?

Response: Thank you for highlighting this ambiguity. We have now rectified this in the revised manuscript by clearly stating that “58%” refers to the percentage of outlier loci identified in the low pH environment that were not found to be outlier loci in the ambient conditions (Lines 186-212). This was simply computed as: (the number of low pH-specific loci/total number of outlier

loci identified in the low pH environment) ($88/151 = 0.58$).

*#3 Reviewer 2 - L231 "As pH tolerance has been shown to exhibit heritability in Mytilus spp.35
"Kingston et al. 2018 (cited in the bibliography as #40) have also shown significant heritability
for calcification rates under OA stress.*

Response: We have added this additional reference in the revised manuscript (Line 309).